



# Deriving the hygroscopicity of ambient particles using low-cost optical particle counters

Wei-Chieh Huang[1], Hui-Ming Hung[1*], Ching-Wei Chu[1], Wei-Chun Hwang[1], and Shih-Chun Candice Lung[2]

[1]Department of Atmospheric Sciences, National Taiwan University, Taipei, 106319, Taiwan
[2] Research Center for Environmental Changes, Academia Sinica, Taipei, 115201, Taiwan

*Correspondence to*: Hui-Ming Hung (hmhung@ntu.edu.tw)

**Abstract.** This study investigates the chemical composition and physical properties of aerosols, which play a crucial role in influencing human health, cloud physics, and local climate. Our focus centers on the hygroscopicity of ambient aerosols, a key

property reflecting the ability to absorb moisture from the atmosphere and serve as cloud condensation nuclei. Employing home-built Air Quality Box (AQB) systems equipped with low-cost sensors, we assess the ambient variability of particulate matter (PM) concentrations to determine PM hygroscopicity. The AQB systems effectively captured meteorological parameters and most pollutant concentrations, with high correlations observed compared to Taiwan Environmental Protection Administration (EPA) data. With the application of κ-Köhler equation and certain assumptions, AQB-monitored PM

concentrations are converted to dry particle mass concentration, showing improved correlation with EPA data and optical particles counter sensitivity correction. The derived κ values range from 0.15 to 0.29 for integrated fine particles ($PM_{2.5}$) and 0.05 to 0.13 for coarse particles ($PM_{2.5-10}$), consistent with results of ionic chromatography analysis for samples from a previous winter campaign nearby. Moreover, the analysis of $PM_{10}$ division into $PM_{2.5}$ and $PM_{2.5-10}$, considering composition heterogeneity, provided improved dry $PM_{10}$ concentration as the sensitivity coefficients for $PM_{2.5-10}$ were notably higher than

for $PM_{2.5}$. Our methodology provides a comprehensive approach to assess ambient aerosol hygroscopicity, offering significant implications for atmospheric modeling, particularly in evaluating aerosol efficiency as cloud condensation nuclei and in radiative transfer calculations. Overall, the AQB systems proved to be effective in monitoring air quality and deriving key aerosol properties, contributing valuable insights into atmospheric science.



# 1 Introduction

In an era of increased industrialization, individuals face growing exposure to poor air quality, elevating the risks of cardiovascular and respiratory diseases (Chen et al., 2017; Brook et al., 2010; Heus et al., 2010). Within the realm of air pollutants, atmospheric aerosols emerge as critical components, playing a vital role in Earth's climate system. They influence radiative balance, cloud formation, and precipitation patterns, while significantly impacting human health, visibility, and ecosystems (Pöschl et al., 2010; Wu et al., 2010; Brook et al., 2010; Hamanaka and Mutlu, 2018). Their ability to scatter and

absorb solar radiation, coupled with their role as cloud condensation nuclei (CCNs), emphasizes their significance in shaping both climate dynamics and air quality (Andreae and Rosenfeld, 2008; Rosenfeld et al., 2014; Lohmann and Feichter, 2005). However, understanding the complex interplay between aerosols and these processes requires the physical and chemical properties of aerosols, including hygroscopicity. The hygroscopic growth of aerosol particles, indicating their ability to absorb moisture from the ambient air, alters their size distribution, mass, optical properties, and CCN activity, thereby impacting

climate dynamics and air quality (Petters and Kreidenweis, 2007). While traditional methods such as hygroscopicity tandem differential mobility analyzers (HTDMA) and cloud condensation nuclei counters (Chan and Chan, 2005; Hung et al., 2016; Bian et al., 2014) have provided valuable insights into the hygroscopic properties of various aerosol types. However, their complexity and cost often limit their applicability for extensive, long-term measurements.

Over the past decade, the rise in popularity of low-cost optical particle counters (OPCs) can be attributed to their simplicity,

portability, and affordability (Sá et al., 2022; Crilley et al., 2018; Samad et al., 2021). OPCs provide real-time data on particle size distributions and mass concentrations with high temporal resolution for monitoring ambient particles. However, challenges arise in ensuring the accuracy of OPCs, necessitating additional constraints or calibrations for optimal performance. The measurement principle of OPCs relies on the dependence of Mie scattering on particle size, yet this dependence is non-monotonic across all sizes. Additionally, particle composition influences light scattering, leading to varying scattering

efficiencies (Kaliszewski et al., 2020; Formenti et al., 2021). Variations in particle density directly affect the mass concentration derived from the monitored number size concentration (Hagan and Kroll, 2020; Dacunto et al., 2015). A particularly challenging issue involves the removal of absorbed liquid water from ambient particles. Several studies have attempted to derive the dry mass concentration of ambient particles using OPC, employing calibration methods linked to the hygroscopic growth factor (HGF) under controlled relative humidity (RH) conditions. Notably, Crilley et al. (2018) improved

OPC mass concentration correction by applying derived $\kappa$ values of 0.38-0.41 and 0.48-0.51 for $PM_{2.5}$ and $PM_{10}$, respectively, achieving a 33 % improvement. Similarly, Antonio et al. (2018) and Jagatha et al. (2021) elevated calibration from a moderate to a high correlation by assuming a constant $\kappa$ of 0.40. Furthermore, the chemical composition and physical properties of aerosols exhibit high temporal-spatial variation, making the analysis and correction of observational data from a physical perspective crucial. The widespread adoption of low-cost sensors, attributed to their affordability, enables more extensive use

as users find them more accessible (Castell et al., 2017). This increased utilization enhances spatial resolution in environmental





monitoring, deepening our understanding of pollution evolution. However, it is essential to emphasize that regular maintenance and calibration are necessary for accurate results (Concas et al., 2021; Sá et al., 2022).

In this study, we evaluate the performance of our home-built monitoring systems through a comprehensive analysis and calibration by co-locating them with the Taiwan Environmental Protection Administration (TW-EPA) station. Our primary
focus is on OPCs, for which we employed a physical model to elucidate the hygroscopic characteristics of ambient particles during the determination of dry particle mass concentrations for integrated fine particles ($D_p \leq 2.5$ μm) and coarse particles ($2.5$ μm $< D_p \leq 10$ μm), respectively. Additionally, we discuss various factors contributing to errors in hygroscopicity estimates, aiming to gain valuable insights into using low-cost sensors for extensive and prolonged monitoring applications.

## 2 Methodology

### 2.1 AQB system

Two home-built AQB systems (AQB #1 and AQB #2) consist of multiple sensors that monitor meteorological parameters such as temperature (T), relative humidity (RH), and pressure (P), as well as gaseous species, and particulate matter (PM) with a temporal resolution of seconds as shown in Fig. 1 with sensor information summarized in Table S1. The gas sensors include five Alphasense amperometric B4 series sensors that measure CO, NO, $NO_2$, Ox ($O_3+NO_2$), and $SO_2$, a photo-ionization
detector (PID-AH2, Alphasense) monitoring volatile organic compounds (VOCs), and a non-dispersive infrared $CO_2$ sensor from Amphenol Advanced Sensors (T6713-5K). The PM sensor (OPC-N2, Alphasense), an optical particle counter, monitors the number size distribution between 0.38 and 17 μm, divided into 16 bins based on Mie scattering, with a sampling flow rate of ~ 4 mL s$^{-1}$ and a refractive index of 1.5+i0. In addition, the mass concentration of $PM_1$, $PM_{2.5}$, and $PM_{10}$ could be calculated from the number size distribution, assuming a particle density of 1.65 g cm$^{-3}$. These sensors were controlled by a small single-
board computer, Raspberry Pi Zero W, at a time resolution of 3 s with data stored in a microSD card and uploaded to cloud storage via 4G LTE. The entire system is housed in a remodeled enclosure measuring 25 cm × 16 cm × 8 cm (L × D × H). The sampling flow rate is controlled by a fan at ~ 5.6 L min$^{-1}$, corresponding to a residence time of approximately 34 s in the box.

### 2.2 Calibration campaign and reference data

The calibration of AQB sensors was carried out by co-locating them with TW-EPA Nanzi station in Kaohsiung, Taiwan
(22°44'12" N, 120°19'42" E) from 4 to 19 February 2021 (Fig. S1). At Nanzi station, the main gaseous components, dry $PM_{2.5}$ and $PM_{10}$ concentrations, and basic meteorological parameters are continuously monitored with instrumentation information summarized in Table S1. For electrochemical sensors, their performance can be influenced by environmental parameters such as temperature, relative humidity, and other chemical species that have high cross-sensitivity (Concas et al., 2021; Karagulian et al., 2019; Mead et al., 2013). Therefore, in this study, a linear regression with a multivariate function of voltage and the
environmental temperature was applied to retrieve concentrations for gas species. For PM, the reported values from the EPA



station (using METONE BAM1020) reflect the dry-state PM concentration by controlling the measurement at RH less than 50 % (i.e., a heating device applied to reduce the sampling flow to 35 % water saturation when the ambient RH is > 50 %). On the contrary, the optical particle counter (OPC) in AQB directly monitors ambient PM concentration. The difference between EPA and AQB reflects the amount of absorbed liquid water in ambient conditions. A simple linear regression between them

might not reveal the influence of hygroscopicity completely. Therefore, the κ-Köhler equation (Petters and Kreidenweis, 2007) was applied to derive the κ as discussed in the following section.

**2.3 Sensitivity coefficients of OPCs and particle hygroscopicity**

To bridge the PM concentration gap between EPA and AQB, the sensitivity correction of OPC and the conversion of ambient particles to dry particles are required. The sensitivity coefficient (α) was evaluated as the ratio of EPA and OPC mass

concentration for data at low RH (≤ 50 %) having limited water content, as follows:

$$\alpha = \frac{M_{EPA}}{M_{OPC}} \tag{1}$$

where $M_{EPA}$ and $M_{OPC}$ are PM concentrations measured by EPA and OPC, respectively. RH ≤ 50 % was applied as the threshold criteria for data selection to determine α, as the mass concentration of ambient particles might have significant water uptake at higher RH. The statistical distribution of all $M_{EPA}$ to $M_{OPC}$ ratios was analyzed to assign α as the mean value ± 0.5σ (σ: standard

deviation) to prevent high-concentration data points from dominating the statistical result.

The particle size growth with the water saturation ratio (S) for a given hygroscopicity (κ) can be evaluated using κ-Köhler equation as follows (Petters and Kreidenweis, 2007):

$$S = \frac{D_{amb}^3 - D_d^3}{D_{amb}^3 - D_d^3(1-\kappa)} exp(\frac{4\sigma_{s/a}M_w}{RT\rho_w D_{amb}}) \tag{2}$$

where $D_{amb}$ and $D_d$ are the diameters of the ambient and dry particulate matter, respectively, $\sigma_{s/a}$ is the surface tension of the

particle, $M_w$ is the molecular weight of water, $R$ is the gas constant, and $\rho_w$ is the density of liquid water. The first term is the solute effect while the second term is the Kelvin effect. As the mass is dominated by the larger particles, the Kelvin effect in Eq. 2 is assumed to be negligible for simplification. The derived dry mass concentration ($M_{d, derived}$) from the measured ambient particles from AQB ($M_{OPC}$) can be expressed as follows (Pope et al., 2010; Crilley et al., 2018):

$$M_{d,derived} = (\alpha \times M_{OPC}) \times \left[\left(\frac{S\kappa}{1-S}\right) \times \frac{\rho_w}{\rho_d} + 1\right]^{-1} \tag{3}$$

where $\alpha$ is the sensitivity coefficient (Eq. 1), $\rho_w$ is the density of liquid water (1.0 g cm⁻³), and $\rho_d$ is the density of dry aerosol particles (assumed to be 1.20 g cm⁻³). With the determined α values (Eq. 1), κ can be derived from the data points of aqueous particles at RH above 70 %, the deliquescence RH (DRH) verified using IC analyzed composition and E-AIM model. The





mean absolute percentage error (MAPE) parameter between $M_{d,derived}$ and $M_{EPA}$ in the following:

$$MAPE = \frac{\sum_{i=1}^{n} \frac{|M_{d,derived,i} - M_{EPA,i}|}{M_{EPA,i}}}{n} \times 100 \text{ %} \tag{4}$$

where $n$ is the total number of data points, was used to assess the appropriate κ value. With the restricted range of α, κ can be derived under the minimum MAPE. Due to the heterogeneity between particles, $PM_{10}$ was divided into integrated fine particles ($D_p \leq 2.5$ μm) and coarse particles (2.5 μm < $D_p \leq 10$ μm) to evaluate the individual sensitivity coefficient and hygroscopicity.

**2.4 Composition analysis**

Hygroscopicity can also be determined using the volume fraction of the major components. Based on an earlier field campaign,
the ion chromatography (IC) method was applied to quantify water soluble components for samples (both $PM_{2.5}$ and $PM_{10}$) collected at Fooyin University (22°36'09.8" N, 120°23'23.1" E) in Kaohsiung from 15 to 28 January 2013. Ambient aerosol samples were collected using a pair of dichotomous aerosol samplers (Model: RP-2025, R&P Co., Inc., Albany, New York) to collect integrated fine and coarse particles on Teflon filters with sampling flow rates of 15.0 and 16.7 L min$^{-1}$, respectively. The samples were categorized into daytime and nighttime. Daytime samples were collected from 08:00 to 20:00 local time
(LT), and nighttime samples were collected from 20:00 to 08:00 LT the next day. The samplers were equipped with Teflon filters deployed for the measurement of water soluble ions ($Na^+$, $Mg^{2+}$, $K^+$, $Ca^{2+}$, $NH_4^+$, $Cl^-$, $SO_4^{2-}$, and $NO_3^-$) via ion chromatography (Model: ICS 1000, Dionex). More information on the chemical analysis method can be found in Salvador and Chou (2014). Additionally, a field campaign conducted in the winter of 2021, focusing only on the analysis of $PM_{2.5}$ was applied to validate the typical hygroscopicity trend in Kaohsiung. We opted for the 2013 dataset due to its comprehensive
analysis encompassing both $PM_{2.5}$ and $PM_{2.5-10}$.

To derive the hygroscopicity from samplings, the ions from IC analysis were converted to chemical components via the following sequence: ammonium sulfate, ammonium bisulfate, ammonium nitrate (when there is residual ammonium), sodium nitrate, and sodium chloride. With the assumption of the hygroscopicity of insoluble components as zero and negligible residual ions contribution (less than 5 % of total mass), the overall hygroscopicity can be derived by the volume fraction ($\varepsilon_i$)
weighted hygroscopicity from individual soluble component ($i$ species) as follows:

$$\kappa = \sum_i \varepsilon_i \kappa_i = \sum_i \frac{v_i}{v_{total}} \kappa_i \tag{5}$$

where $\kappa_i$ is the hygroscopicity of $i$ species, $v_{total}$ is the volume of particles, and $v_i$ is the volume of $i$ species. The conversion of particle mass to volume is based on a density of 1.20 g cm$^{-3}$. The applied hygroscopicity, molecular weight, and density for the related chemical species are summarized in Table S2. With the assumption that these ions dissolve completely in the
aqueous phase and assuming a van't Hoff factor of 1.0, which represents the maximum estimation, the hygroscopic contributed by the residual ions were found to be approximately up to 1.8 % and 6.4 % of the overall κ value for $PM_{2.5}$ and $PM_{2.5-10}$,





respectively. Given their limited impact on the hygroscopic behavior of the particles, the contribution of the residual ions was not taken into account in the calculation. Furthermore, the composition data obtained from IC analysis was used in the Extended Aerosol Inorganics Model (E-AIM) Model III (for systems containing $H^+$, $NH_4^+$, $Na^+$, $SO_4^{2}$, $NO_3^-$, $Cl^-$, and $H_2O$) to evaluate

the characteristics of volume variation as a function of RH in the range of 30 to 90 % (Clegg et al., 1998). The partitioning of selected trace gases ($HNO_3$, $HCl$, $NH_3$, and $H_2SO_4$) into the vapor phase was disabled to keep a consistent quantity of applied chemical species in the particle phase. The growth factor, $V_{amb}/V_d$, above DRH, was applied to retrieve κ value using Eq. 2 but without the Kelvin effect term (Luo et al., 2020). Both the individual sample concentrations and the overall average conditions were analyzed to evaluate the hygroscopic behavior of the particles.

## 3 Results and Discussion

### 3.1 Performance of AQB systems

Figure 2 shows the time series of the meteorological parameters and pollutant concentrations between calibrated AQB and EPA data from 14 to 17 February 2021. T, RH, CO, and Ox showed a good correlation with r > 0.9, while NO, $NO_2$, $PM_{2.5}$, and $PM_{10}$ had a moderate correlation (r ≥ 0.48). The NMHC sensor can only detect the peaks of high concentrations and cannot

reveal temporal variation at low concentrations, resulting in a low correlation. Overall, the AQB system performs well in capturing the ambient variability of pollutants stated above. The low correlation of $SO_2$ was due to the cross-sensitivity of this $SO_2$ sensor, which was highly sensitive to $O_3$ and $NO_2$ (about -120 % reposted in the Technical Specification of Alphasense). $SO_2$ generally has lower concentrations than $O_3$ and $NO_2$, which dominate the response of the $SO_2$ sensor. However, if high $SO_2$ concentration events occur, the $SO_2$ sensor might reflect the variation of $SO_2$ concentration. The PM concentration in Fig.

2 was calibrated by a simple linear regression and could reflect the trend of mass concentration roughly but with a more significant deviation at higher RH due to the additional absorbed water, which is discussed in section 3.2. Most gas species showed a high correlation (r ≥ 0.95) between different AQB systems except for NMHC (r = 0.675) as summarized in Table S3. Further results and discussions focus on the PM analysis using AQB #1, which has a more consistent sampling rate during the observation period, unless stated otherwise.

### 3.2 Derived Hygroscopicity

Figures 3(a) and 3(b) show the scatter distribution of the mass concentration between AQB #1 (with no calibration) and EPA data for $PM_{2.5}$ and $PM_{10}$, respectively. Overall, the PM mass concentration measured by AQB system appears to be higher than that measured by EPA. The results reveal a clear correlation between ambient RH and the ratio of ambient particles to dry particles, indicating the contribution of water content. The red-shaded area represents a regression line with a slope

corresponding to the inverse of the sensitivity coefficients (α) derived from data points at ambient RH ≤ 50 %. The significant deviation of the red shaded area from the 1:1 line towards the right side indicates the requirement of α > 1 correction, contributed by the different measurement principles and calibration techniques, which may result from the assuming a particle





density and refractive index (RI) (dust, density: 1.65 g cm$^{-3}$, RI: 1.5 + i0). The estimated α, summarized in Table 1, are higher for PM$_{10}$ than for PM$_{2.5}$, i.e. 2.02 ± 0.34 vs 1.26 ± 0.16. The deviation in α might be attributed to the complex composition of

ambient particles, which differs from the samples used for instrument calibration, as well as possible sensitivity variations in OPC over time. The performance in ambient RH ≤50 % exhibits a strong correlation with coefficient of determination ($R^2$) at 0.98 for PM$_{2.5}$ and 0.90 for PM$_{10}$, respectively. The correlation performance is similar to other real-time outdoor field studies reporting $R^2$ ranging from 0.79 to 0.99 for PM$_{2.5}$ and 0.82 to 0.84 for PM$_{10}$ (Gillooly et al., 2019; Demanega et al., 2021; Sá et al., 2022; Crilley et al., 2018). Additionally, the OPC sampling flow rate has an impact on measured performance. For AQB

#1, the sampling flow rate remains relatively steady at 3.6±0.2 LPM. In contrast, AQB #2 exhibits two distinct time periods with sampling flow rates of 3.6-4.2 LPM for the first period and 3.2-3.6 LPM for the second period. The distinctive sampling flow rates result in a non-linear change in α, suggesting the need to separate the data into two parts to estimate the individual α (see Fig. S2).

With the derived α, the hygroscopicities can be retrieved using Eq. (3), ranging from 0.18 to 0.29 for PM$_{2.5}$ and 0.20 to 0.38

for PM$_{10}$ (Table 1). The results obtained from the two AQB systems exhibit slight differences but are consistent. Considering the sensitivity coefficient and hygroscopicity, Figures 3(c) and 3(d) show the scatter distribution of the derived dry concentration vs. EPA concentration under the lowest MAPE for PM$_{2.5}$ and PM$_{10}$, respectively. However, due to the heterogeneity of composition among different sizes, PM$_{10}$ can be divided into integrated fine particles (PM$_{2.5}$) and coarse particles (PM$_{2.5-10}$, 2.5 μm < D$_p$ ≤ 10 μm) for further analysis. The estimated α value for PM$_{2.5-10}$, as summarized in Table 1, is

approximately one order of magnitude higher than that for PM$_{2.5}$. Figure 3(e) shows the scatter distribution between the derived dry PM$_{2.5-10}$ from AQB data and EPA data, exhibiting a MAPE of 31.8 %, more significant than the 24.8 % for PM$_{2.5}$. The higher MAPE might result from the low particle number concentration in the coarse mode, with only about 0.01 to 0.1 particles per bin cm$^{-3}$ in the size range of 3.0 to 10.0 μm. Detection efficiency may be influenced by notable spatial variations. This observation aligns with the findings reported in the study by Kaliszewski et al. (2020), which showed a reduced correlation

between OPC-N3 measurements and reference instruments for larger particles. The dry PM$_{10}$ derived from AQB through the divided PM$_{2.5}$ and PM$_{2.5-10}$ analysis demonstrates a better consistency with the reported EPA data than the direct calibration method, i.e., a lower MAPE in Fig. 3(f) than that in Fig. 3(d). The derived κ for PM$_{2.5-10}$ is 0.07 − 0.13, lower than that of PM$_{2.5}$ (0.18-0.29). The lower κ for PM$_{2.5-10}$ might suggest a significant contribution from dust or other less hygroscopic species, aligning with the IC analysis in Table 2 and discussed further in section 3.3.

**3.3 Aerosol Composition and E-AIM Model**

The major soluble composition and concentrations obtained from the IC analysis are summarized in Table 2, showing mean PM$_{2.5}$ and PM$_{2.5-10}$ concentrations of 67±19 and 36±7 μg m$^{-3}$, respectively. The determined PM$_{2.5}$ soluble composition constitutes approximately 53 % of the mass fraction and is predominantly composed of NH$_4^+$, SO$_4^{2-}$, and NO$_3^-$. These components are formed through chemical reactions involving industrial and agricultural emissions. In contrast, PM$_{2.5-10}$





exhibits ~ 30 % of soluble components, including $NO_3^-$, $SO_4^{2-}$, $Na^+$, $Cl^-$, $NH_4^+$, and some alkaline earth metal ions ($Ca^{2+}$ and $Mg^{2+}$) and a larger proportion of insoluble components (~70 %), likely attributed to dust, metallic components, and unanalyzed organic-components. The increased sea salt content ($Na^+$ and $Cl^-$) is likely transported by the sea breeze in the daytime, while the increased fractions of $Ca^{2+}$ and $Mg^{2+}$ might correspond to sand or dust particles (Li et al., 2022). The temporal variation of derived $\kappa$, based on the IC soluble composition analysis, ranges from 0.14 to 0.26 for $PM_{2.5}$ and 0.06 to 0.21 for $PM_{2.5-10}$, as

shown in Fig. S3(a) and summarized in Table 1. The obtained $\kappa$ value for $PM_{2.5}$ is consistent with that derived from data in the winter of 2021, as illustrated in Fig. S4. This consistency highlights the reliability of our findings, demonstrating the robustness across distinct study periods. The more significant variability in $\kappa$ for $PM_{2.5-10}$ compared to $PM_{2.5}$ can be attributed to the pronounced fluctuations in the soluble composition of coarse particles, primarily driven by substantial quantities of thenardite ($Na_2SO_4$) and halite (NaCl) (Tang et al., 2019). Due to the dominance of the northeast monsoon wind during the filter sampling

period, the influence of the sea-land breeze was relatively weak to cause apparent diurnal variation in $\kappa$. The derived $\kappa$ value for $PM_{2.5}$ from IC analysis (0.14-0.27) are consistent with that obtained from AQB analysis (~0.22), while the $\kappa$ value for $PM_{2.5-10}$ from IC analysis (0.06-0.21) is relatively higher than that from AQB analysis (~0.08) (Table 1 and Fig. 4(a)). The differences in $\kappa$ between the IC and AQB analyses could be attributed to the spatial and temporal variations in aerosols, as well as the different campaign years and locations (~20 km apart, as shown in Fig. S1). These differences might also be

influenced by technique uncertainties, such as ammonia and nitrate sampling evaporation during filter sampling (Hering and Cass, 1999; Chen et al., 2021), as well as OPC detection uncertainties. Overall, the derived $\kappa$ values from the OPC data in AQB likely reflect the hygroscopicity of both integrated fine and coarse particles.

The particle growth might follow the $\kappa$-Köhler equation (Eq. 2) when all soluble species are fully dissolved, typically occurring above the DRH. With the averaged soluble composition determined from the IC analysis, HGF as a function of RH calculated

using E-AIM is shown in Fig. 5. For $PM_{2.5}$, partial deliquescence initiates at 60 % of RH resulting in residual solid components such as (($NH_4)_2SO_4$ and $2NH_4NO_3.(NH_4)_2SO_4$). Complete dissolution occurs around an RH of 72 % as the DRH. In the case of $PM_{2.5-10}$, water uptake begins at RH = 42 %, leaving a residual solid composed of $3NH_4NO_3(NH_4)_2SO_4$, $NH_4Cl$, and $NaNO_3.Na_2SO_4.H_2O$ until reaching RH of 68 %. The daily DRH happens at 71.3±4.9 and 67.1±3.4 for $PM_{2.5}$ and $PM_{2.5-10}$, respectively, as shown in Figs. S3(b) and S3(c). In the AQB data analysis, a threshold of RH ≥ 70 % was applied to ensure

sufficient data points but slightly lower than DRH of $PM_{2.5}$. To assess the potential bias associated with the selected DRH threshold, Fig. S5 shows the HGF of mean soluble composition as a function of RH estimated using E-AIM. With Eq. 2 (without the Kelvin effect term) and the assumption of volume additivity between particle and updated water, $\kappa$ derived using 70 % and 75 % thresholds show less than 1% of the difference for both integrated fine and coarse compositions, but 13 % and 6 % less than that estimated from the composition of $PM_{2.5}$ and $PM_{2.5-10}$, respectively. The $\kappa$ deviation by the applied threshold

appears negligible in this studied condition. The performance is similar to that obtained from the analysis of AQB data. As the threshold becomes smaller, the derived $\kappa$ decreases slightly but with a broader uncertainty (Fig. S6). However, the temporal composition variation for the applied AQB data set (~ 16 days of observation) might lead to a higher variation. Furthermore, the 13 % lower $\kappa$ for E-AIM than the composition estimation is likely due to the RH-dependent ionic activities following the



Zdanovskii-Stokes-Robinson relation in E-AIM. The calculation based on Eq. 2, with volume additivity assumptions, might
overestimate the liquid water content. Similar findings were reported by Kreidenweis et al. (2008) regarding the percentage
difference between κ-Köhler equation- and E-AIM-derived water contents increasing with RH. Overall, κ derived from the
growth profile might be smaller than the composition estimation (associated with the cloud nuclei activation), likely due to the
assumptions of volume additivity and the fixed van't Hoff factor in the κ-Köhler equation.

**3.4 Uncertainty Discussion**

For simplicity, we derived κ from AQB data without considering the Kelvin effect and under an assumed particle density. The
ignorance of the Kelvin effect might result in minor differences for particles larger than 100 nm under sub-saturated conditions
(Pope et al., 2010; Topping et al., 2005; Crilley et al., 2018). To confirm the appropriateness, we assessed biases for particles
at 0.1 and 1 μm without considering the Kelvin effect, as shown in Fig. S7. For particles with a κ value of 0.3 under RH ranging
from 70 to 95 %, the deviation of κ due to neglecting the Kelvin effect is -10 % for 0.1 μm particles and -1 % for 1 μm particles,
decreasing with particle diameter. The growing particle diameter is overestimated under the same RH conditions because the
positive Kelvin effect is ignored. To compensate for the deficiency in particle saturation, the balanced particle diameter needs
to be more significant with a larger solute effect. However, the average mass-weighted mean diameter for $PM_{2.5}$ is about 1.3
μm. Therefore, the ignorance of Kelvin's effect on the analysis might have limited influence on the derived κ. This phenomenon
becomes more significant with increasing RH, resulting in a more considerable underestimation of κ values under high RH
conditions. During our monitoring campaign, the surrounding RH ranged from 31 to 92 %, and we focused on deriving κ
values for integrated fine and coarse particles. Therefore, the assumption of a negligible Kelvin effect is proper for this study.
Furthermore, the derived κ using Eq. (3) for AQB data or Eq. (5) for IC data is notably influenced by the assumed particle
density. Assuming that the undetermined composition mainly consists of secondary organic species, having a density of 1.2 g
$cm^{-3}$, within the reported densities ranging from 0.9 to 1.6 g $cm^{-3}$ depending on the formation process (Malloy et al., 2009;
Kostenidou et al., 2007; Zelenyuk et al., 2008), along with the analyzed soluble chemical species summarized in Table S2, the
calculated densities for $PM_{2.5}$ and $PM_{2.5-10}$ are 1.42±0.03 and 1.34±0.07 g $cm^{-3}$, respectively (Fig. S8). This increases densities
by about 15 % and 10 % for $PM_{2.5}$ and $PM_{2.5-10}$, respectively. The derived κ from AQB data increases by approximately 17 %
and 9 % for $PM_{2.5}$ and $PM_{2.5-10}$, respectively, while the derived κ from IC data is proportional to density (15 % and 10 % for
$PM_{2.5}$ and $PM_{2.5-10}$, respectively) as shown in Fig. 4(b). Overall, the derived κ exhibits consistency between the AQB and IC
analysis. This bias might be intensified if components having a higher portion of composition with larger density, such as
black carbon (a non-hygroscopic species with κ ~ 0) having a high density of about 1.8 g $cm^{-3}$ (Park et al., 2004; Shiraiwa et
al., 2008) are taken into consideration.





## 4 Conclusion

In this study, we evaluated the performances of home-built Air Quality Box (AQB) systems equipped with low-cost sensors
and focused on the ambient variability of particulate matter (PM) concentrations to derive the hygroscopicity of PM. The AQB
systems revealed their effectiveness in capturing meteorological parameters and most pollutant concentrations. Notably,
compared to EPA data, high correlations were observed for parameters such as temperature, relative humidity, CO, and Ox
($O_3 + NO_2$) (r ≥ 0.96). While NOx and PM exhibited moderate correlations (r ≥ 0.48), the NMHC sensor showed limitations
in capturing temporal variations at low concentrations, and the $SO_2$ sensor faced cross-sensitivity challenges. Calibration of
PM concentration through linear regression demonstrated general agreement with EPA data, although deviations at higher
relative humidity indicated the influence of absorbed water. Applying the κ-Köhler equation and assuming constant particle
density, AQB-monitored PM concentration can be converted to dry particle mass concentration, aligning well with EPA data
after OPC sensitivity correction. The derived hygroscopicity provides the relationship between ambient relative humidity and
particle water content. By dividing $PM_{10}$ into $PM_{2.5}$ and $PM_{2.5-10}$, considering the composition heterogeneity, we achieved more
precise dry $PM_{10}$ concentrations with lower MAPE. The sensitivity coefficients (α) for $PM_{2.5-10}$ (10.58 ~ 12.37) were higher
than for $PM_{2.5}$ (1.26 ~ 1.44), reflecting different measurement and calibration approaches. The higher α in the coarse mode
indicated that the detection efficiency may be influenced by notable spatial variations with the low particle number
concentration. The derived κ from AQB data, ranging from 0.15 to 0.29 for $PM_{2.5}$ and 0.05 to 0.13 for $PM_{2.5-10}$, showed
consistent with those from IC soluble composition analysis (0.14 to 0.27 for $PM_{2.5}$ and 0.06 to 0.21 for $PM_{2.5-10}$). Variations in
IC analysis were primarily influenced by the proportion of soluble components, higher in $PM_{2.5}$ (~53 %) than in $PM_{2.5-10}$ (~30
%). The lower κ for $PM_{2.5-10}$ than $PM_{2.5}$ might suggest a significant contribution from dust or other less hygroscopic species.
Our analysis also considered the effects of chosen deliquescence relative humidity (DRH) thresholds and Kelvin effects, which
were found to have a minor impact on underestimating κ values (less than 1 %). Conversely, recalculating particle densities
for $PM_{2.5}$ (1.42±0.03 g cm$^{-3}$) and $PM_{2.5-10}$ (1.34±0.07 g cm$^{-3}$) led to an increase in the derived κ by approximately 17 % and 9
%, respectively, compared to the initial assumption of 1.2 g cm$^{-3}$. Overall, the AQB systems proved effective in monitoring
pollutant concentrations and deriving hygroscopicity, providing valuable data for understanding air quality dynamics. The
method to assess low-cost sensors near EPA stations, might enhance our understanding of the temporal and spatial variability
of aerosol hygroscopicity. The study also emphasizes the need for careful consideration of uncertainties and calibration
techniques for accurate interpretation of AQB data in atmospheric research.

**Code & Data availability**

The code is not publicly accessible, but readers can contact HM Hung (hmhung@ntu.edu.tw) for more information. The
observation data for AQBs and TW-EPA, the E-AIM model output, and the hygroscopicity deriving result used in this study
can be accessed online at https://github.com/NTUACLab/Wei-Chieh.





## Author contributions

WC Huang carried out the calibration campaign, did data analysis, and prepared the manuscript draft. HM Hung supervised the project, including data discussion and manuscript editing. CW Chu and WC Hwang designed the home-built AQB system and did database generation. SCC Lung supervised the field study of 2013, and carried out the aerosol composition analysis in 2021.

## Competing interests

The authors declare that they have no conflict of interest.

## Acknowledgments

We appreciate Taiwan EPA for providing the minute-averaged data of meteorological parameters and chemical species for calibration and comparison and Dr. Shih-Chieh Hsu at Research Center for Environmental Changes, Academia Sinica, Taipei for composition data of $PM_{2.5}$ and $PM_{10}$ in Kaohsiung (2013). This study was supported by the National Science and Technology Council in Taiwan (111-2111-M-002-009 and 112-2111-M-002-014).

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





**Tables**

**Table 1: The sensitivity coefficients and the hygroscopicity for PM$_{2.5}$, PM$_{10}$, and PM$_{2.5-10}$, respectively.**

| | Sensitivity coefficient ($\alpha$) | | Hygroscopicity ($\kappa$) | | | |
| --- | --- | --- | --- | --- | --- | --- |
| | AQB #1 | AQB #2* | AQB #1 | AQB #2 | IC (species) | IC (E-AIM) |
| PM$_{2.5}$ | $1.26 \pm 0.16$ | $1.44 \pm 0.20$ | $0.18 - 0.29$ | $0.15 - 0.24$ | $0.14 - 0.27$ | $0.14 - 0.26$ |
| PM$_{10}$ | $2.02 \pm 0.34$ | $2.20 \pm 0.38$ | $0.20 - 0.39$ | $0.18 - 0.30$ | | |
| PM$_{2.5-10}$ | $12.37 \pm 1.33$ | $10.58 \pm 2.90$ | $0.07 - 0.13$ | $0.05 - 0.09$ | $0.06 - 0.21$ | $0.08 - 0.21$ |

**\* the sensitivity of AQB #2 presents the value in the period of sampling flow rates at 3.6-4.2 LPM**





**Table 2. The total mass concentration, the major water soluble composition and concentration (mean value and standard deviation in µg m$^{-3}$) of winter PM$_{2.5}$ and PM$_{2.5-10}$ in Kaohsiung by ion chromatography. (others presented the insoluble composition)**

| Ion species | Total | Na$^+$ | Mg$^{2+}$ | K$^+$ | Ca$^{2+}$ | NH$_4^+$ | Cl$^-$ | SO$_4^{2-}$ | NO$_3^-$ | others |
|---|---|---|---|---|---|---|---|---|---|---|
| **PM$_{2.5}$** | 67.0 ± 19.2 | 0.31 ± 0.14 | 0.06 ± 0.02 | 0.45 ± 0.14 | 0.08 ± 0.04 | 8.24 ± 2.68 | 1.21 ± 0.91 | 13.63 ± 4.72 | 11.89 ± 4.88 | 31.1 ± 8.0 |
| **PM$_{2.5-10}$** | 36.8 ± 7.64 | 1.50 ± 0.52 | 0.21 ± 0.06 | 0.04 ± 0.02 | 0.74 ± 0.25 | 1.07 ± 0.69 | 1.28 ± 0.69 | 1.87 ± 1.12 | 4.35 ± 1.41 | 25.7 ± 6.4 |




**Figures**

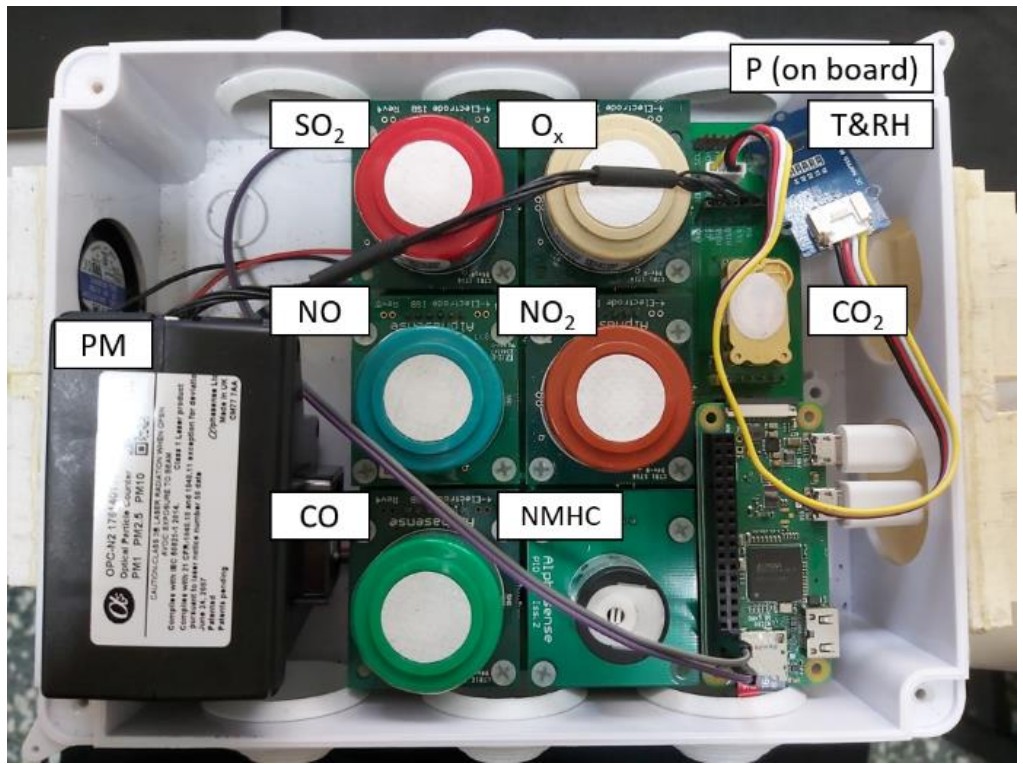

**Figure 1: The design of the AQB system.**






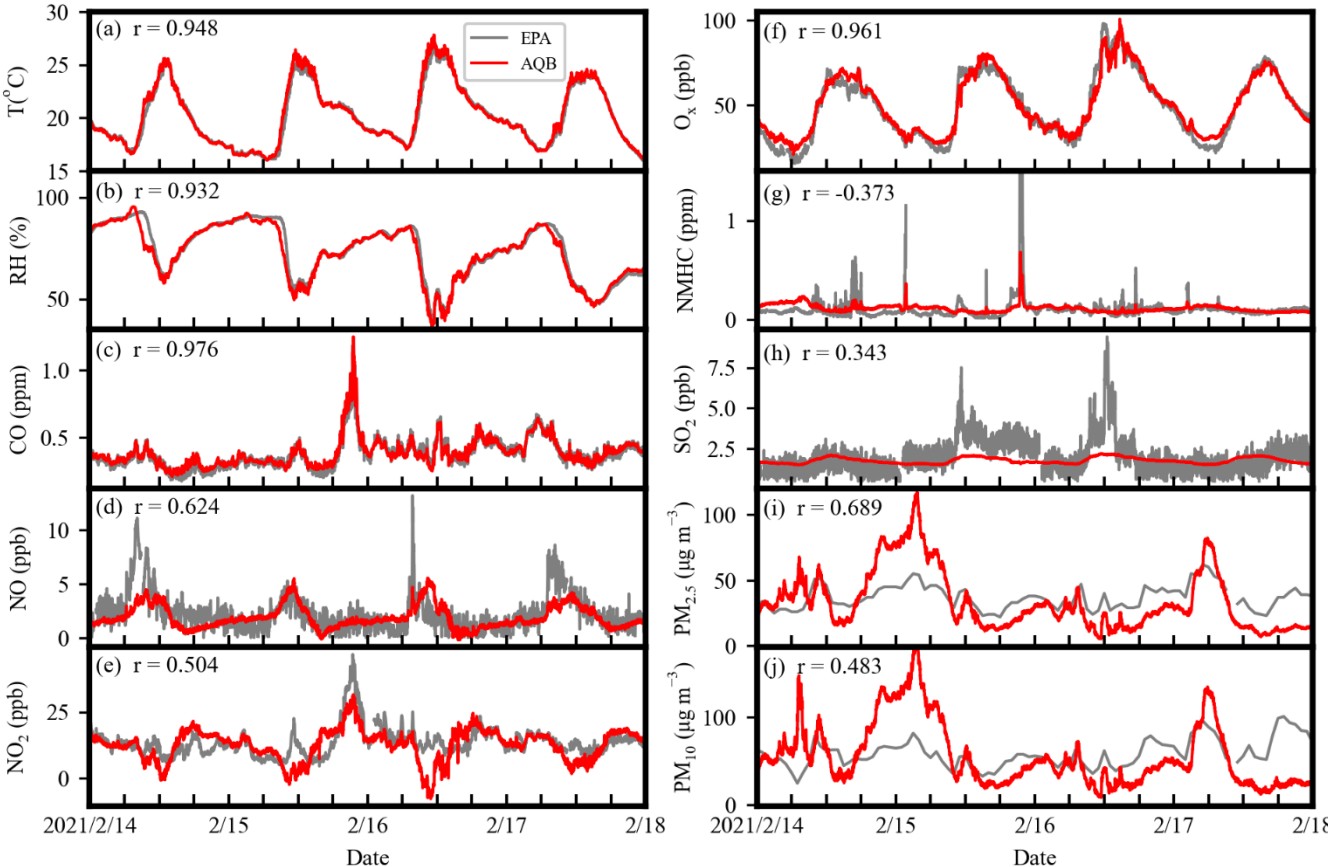

**Figure 2: The temporal profiles of calibrated AQB data (red lines) and the EPA measurement (grey lines) for (a) temperature, (b) relative humidity, (c) CO, (d) NO, (e) NO₂, (f) Ox (≡ NO₂ + O₃), (g) Non-methane hydrocarbon, (h) SO₂, (i) PM₂.₅, and (j) PM₁₀ during the period of 14 – 17 February 2021. All the species were calibrated using linear regression.**




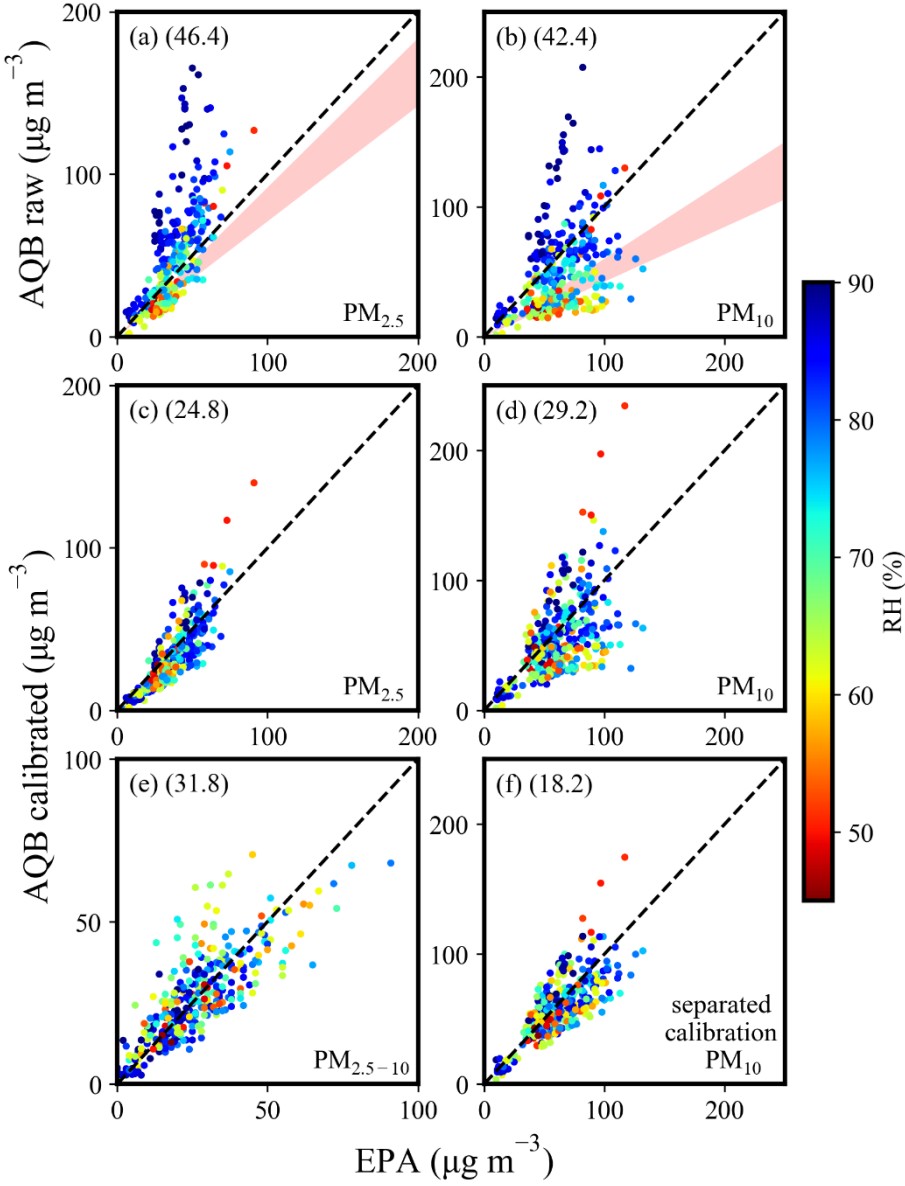

**Figure 3: The correlation of mass concentration between EPA and OPC in AQB #1 (raw data or calibrated data): (a, c) PM$_{2.5}$, (b, d) PM$_{10}$, (e) PM$_{2.5-10}$, and (f) separated calibration PM$_{10}$, respectively. (a-b) are the raw data, while (c-f) are the calibrated data. Marker color corresponds to relative humidity. The shading is the sensitivity coefficient ("α"). The value in parentheses is the**
**MAPE in percentage.**



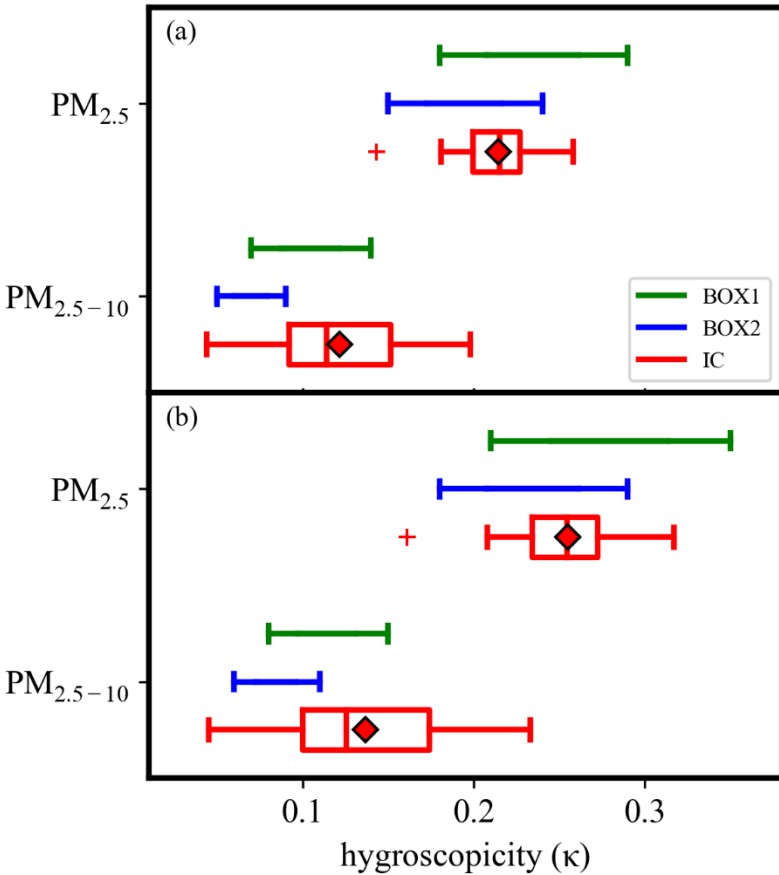

**Figure 4: The hygroscopicities of PM$_{2.5}$ and PM$_{2.5-10}$ derived based on data from AQBs and ion chromatography with the assumption particle density of (a) 1.2 g cm$^{-3}$ and (b) 1.42±0.03 and 1.34±0.07 g cm$^{-3}$, respectively, for PM$_{2.5}$ and PM$_{2.5-10}$ based on partitioning analyzed results in Kaohsiung. Average is shown as a red diamond.**





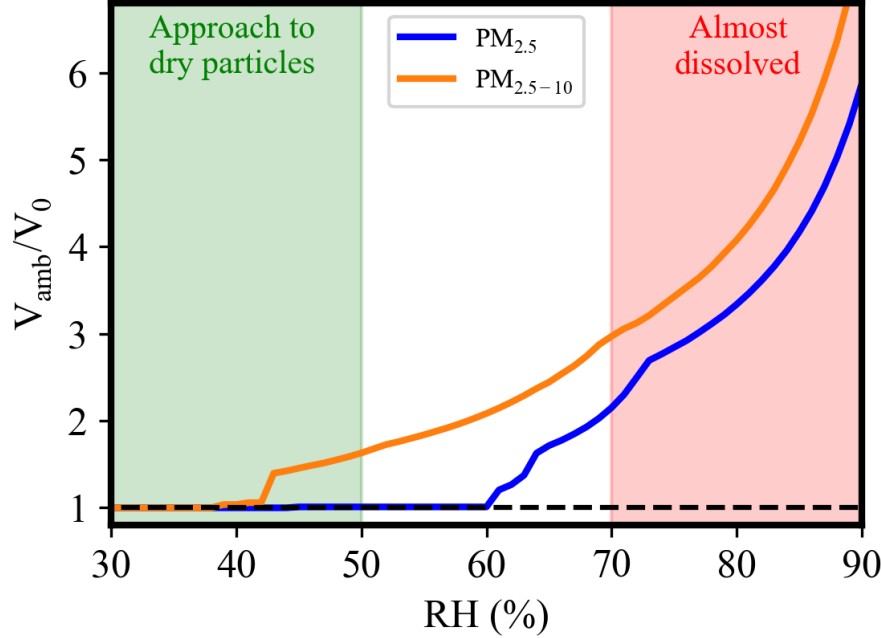

**Figure 5:** **The volume ratio variation of a given soluble composition as a function of RH with thermodynamic equilibrium using E-AIM at 298.15 K. (composition is the averaged IC data with a molarity ratio of Na$^+$:NH$_4^+$:Cl$^-$:SO$_4^{2-}$:NO$_3^-$ as 14:458:0:142:188 for PM$_{2.5}$, and 65:59:16:19:70 for PM$_{2.5-10}$.)**
