# Peer review of "Deriving the hygroscopicity of ambient particles using low-cost optical particle counters"

_Atmospheric Measurement Techniques, 2024_

## Author Response (AR1)

**Response to Reviewers for "Deriving the hygroscopicity of ambient particles using low-cost optical particle counters".**

We would like to thank the anonymous reviewers for the comments that significantly improved the clarity and readability of the manuscript. Our point-by-point responses are found below in blue ink. The revised content is highlighted in yellow.

**RC1**

1. You stated that "AQB-monitored PM concentration can be converted to dry particle mass concentration, aligning well with EPA data after OPC sensitivity correction. The derived hygroscopicity provides the relationship between ambient relative humidity and particle water content. By dividing $PM_{10}$ into $PM_{2.5}$ and $PM_{2.5-10}$, considering the composition heterogeneity, we achieved more precise dry $PM_{10}$ concentrations with lower MAPE." Please state differences clearly with 2 different observing packages.

A: Thank you for your constructive comments and for emphasizing the need for clarity in differentiating between the two observational packages used in our study. The EPA stations (using METONE BAM1020) report dry-state PM concentrations by controlling the measurement environment to maintain relative humidity (RH) below 50%. In contrast, the optical particle counter (OPC) in the AQB directly monitors ambient PM concentrations. The comparison of these two datasets illustrates the sensitivity variation of low-cost OPC sensors and the influence of hygroscopicity. The content of Section 4 (Lines 288-294) was rewritten to clarify this issue as follows: " In the PM analysis, $PM_{10}$ was divided into $PM_{2.5}$ and $PM_{2.5-10}$ to account for compositional heterogeneity among different particle sizes. Comparing the AQB-monitored ambient PM data and the TW-EPA data (for dry particles) at RH ≤ 50%, the derived sensitivity coefficients (α) for $PM_{2.5-10}$ (10.58 - 12.37) were higher than those for $PM_{2.5}$ (1.26 - 1.44) likely due to the significant sensitivity variation in the OPC over time. By considering hygroscopicity with the κ-Köhler equation and assuming a constant composition density for sensitivity-corrected AQB data, the derived dry particle mass concentrations show improved consistency with TW-EPA data compared to the simple linear regression approach."

2. Provide units for the parameters in the equations.

A: Thank you for your kind reminder. All equations are labeled with units to ensure correct use. The adjusted revisions for the descriptions of equations (Eqs.1 and 2) in the paragraph as follows: "

$$\alpha = \frac{M_{EPA}}{M_{OPC}} \qquad (1)$$

where $M_{EPA}$ and $M_{OPC}$ are PM concentrations ($\mu g\ m^{-3}$) measured by TW-EPA and OPC, respectively.

$$S = \frac{D_{amb}^3 - D_d^3}{D_{amb}^3 - D_d^3(1-\kappa)} exp(\frac{4\sigma_{s/a}M_w}{RT\rho_w D_{amb}}) \qquad (2)$$

where $D_{amb}$ and $D_d$ are the diameters (m) of the ambient and dry particulate matter, respectively, $\sigma_{s/a}$ is the surface tension of the particle (J m$^{-2}$), $M_w$ is the molecular weight of water (g mole$^{-1}$), $R$ is the gas constant (J mole$^{-1}$ K$^{-1}$), and $\rho_w$ is the density of liquid water (1.0 g m$^{-3}$). The first term is the solute effect, while the second term is the Kelvin effect."

In Eqs.3 and 4, units for the parameters in the equations have already been provided in Eqs. 1 and 2 or the paragraph. For Eq.5, the volume mixing ratio ($\varepsilon$) and hygroscopicity ($\kappa$) is a dimensionless quantity.

3. How accurate your hygroscopicity calculation that needs to be discussed.

A: In this study, the low-cost sensors acquired data for a certain period to cover a more comprehensive RH range for the hygroscopicity calculation. In the studied case, the analysis successfully showed different mean $\kappa$ values between PM$_{2.5}$ and PM$_{2.5-10}$ based on the data of the monitored period. Even though there was no intensive filter sample collection during the studied period, the comparison of two IC analysis results from other studies (2013 and 2021 winter campaigns) shows similar hygroscopicity of PM$_{2.5,}$ ranging from 0.18 to 0.25, which might represent a typical winter PM$_{2.5}$ hygroscopic characteristics in Kaohsiung city. The consistent results between the two winter campaigns might suggest the overall soluble species fraction in PM$_{2.5}$ is generally within a similar range. Therefore, the derived mean $\kappa$ range under the specific assumption (solute density and ignorance of Kelvin effect) from AQB data was discussed for the range consistency compared to the results of available two winter campaigns for PM$_{2.5}$ and the 2013 winter campaign for PM$_{2.5-10}$. However, the accuracy improvement of dry particle mass concentration derived from the AQB monitored data after applying the OPC sensitivity coefficient and derived $\kappa$ values can be evaluated as illustrated in Section 3.2 and summarized in a new table (Table 2).

Table 2: Performance metrics of different calibration methods for PM$_{2.5}$, PM$_{2.5-10}$, and PM$_{10}$.

| | PM$_{2.5}$ | | | PM$_{2.5-10}$ | | | PM$_{10}$ | | | |
|---|---|---|---|---|---|---|---|---|---|---|
| | RH≤50% Only[a] | All data (no κ) | All data (κ= 0.29) | RH≤50% Only[a] | All data (no κ) | All data (κ= 0.09) | RH≤50% Only[a] | All data (no κ) | All data (κ= 0.36) | (PM$_{2.5}$+ PM$_{2.5-10}$)[c] |
| applied α | 1.26±0.16 | 1.04 | 1.40 | 12.37±1.33 | 10.77 | 13.16 | 2.02±0.34 | 1.69 | 2.36 | — |
| MAPE (%) | 21.3 (12.8) | 48.8 | 24.8 | 15.9 (11.5) | 37.9 | 31.8 | 32.8 (18.5) | 62.5 | 29.2 | 18.2 |
| RMSE (μg cm$^{-3}$) | 20.5 (3.7) | 29.1 | 11.3 | 4.9 (2.8) | 9.4 | 9.1 | 42.6 (10.3) | 54.7 | 26.9 | 15.9 |
| R$^2$ [b] | .0.55 (0.51) | -3.49 | 0.32 | 0.31 (0.78) | 0.57 | 0.59 | -4.18 (-0.58) | -4.74 | -0.38 | 0.51 |

[a] Only for data points at RH ≤50%. The value in parentheses is the performance result without two significant outliers shown in Fig. 3

[b] Coefficient of determination (R$^2$) is calculated as the proportion of variation in the calibrated dry mass concentration.

[c] The combination of calibrated data from PM$_{2.5}$ All data (κ= 0.29) and PM$_{2.5-10}$ All data (κ= 0.09).

To clarify this approach, Sections 3.2 and 3.3 were revised as follows: in Section 3.2 (Lines 172-211): "Figures 3(a) and 3(c) show the scatter distribution of the mass concentrations between AQB #1 (with no calibration) and TW-EPA data for $PM_{2.5}$ and $PM_{10}$, respectively. Overall, the PM mass concentrations measured by AQB system appear to be higher than those reported by TW-EPA. The results reveal an apparent influence of ambient RH, indicating the contribution of water content. The red-shaded area represents a regression line with a slope corresponding to the inverse of the sensitivity coefficients ($\alpha$) derived from data points at ambient RH $\leq$ 50% (17 out of 356 points, 5%). The notable deviation of the red shaded area from the 1:1 line towards the right side indicates the requirement of $\alpha > 1$ corrections, contributed by the different measurement principles and calibration techniques, which may result from the assuming particle density and refractive index (RI) (dust, density: 1.65 g $cm^{-3}$, RI: 1.5 + 0i). The estimated $\alpha$, as summarized in Table 1, are higher for $PM_{10}$ than for $PM_{2.5}$, i.e., 2.02 $\pm$ 0.34 vs 1.26 $\pm$ 0.16, which are reasonably conclusive as tested with more data points selected at higher RH thresholds (Fig. S2). The $\alpha$ difference between $PM_{2.5}$ and $PM_{10}$ might be attributed to the complex composition of ambient particles, which differs from the samples used for instrument calibration, as well as possible sensitivity variations in OPC over time. With sensitivity calibration, the performance at ambient RH $\leq$ 50% exhibits a strong correlation with MAPE at 12.8%, 18.5%, and Root Mean Squared Error (RMSE) at 3.7 $\mu$g $m^{-3}$, 10.3 $\mu$g $m^{-3}$ for $PM_{2.5}$ and $PM_{10}$, respectively, as summarized in Table 2 excluding the two significant outliers (shown as hollow circles in Fig. 3). The results confirm the effectiveness of OPCs in capturing PM concentrations, consistent with previous real-time outdoor field studies (Gillooly et al., 2019; Demanega et al., 2021; Sá et al., 2022; Crilley et al., 2018). Additionally, the OPC sampling flow rate has an impact on measurement performance. AQB #1 maintained a steady rate at 3.6 $\pm$ 0.2 LPM, whereas AQB #2 exhibits two distinct time periods with sampling flow rates of 3.6-4.2 LPM for the first period and 3.2-3.6 LPM for the second period. …With the derived $\alpha$, the hygroscopicities were retrieved using Eq. (3), resulting in $\kappa$ ranging from 0.18 to 0.29 for $PM_{2.5}$ and 0.20 to 0.39 for $PM_{10}$ (Table 1) during the studied period. Figures 3(d) and 3(f) show the scatter distribution of the derived dry concentration vs. TW-EPA concentration for $PM_{2.5}$ and $PM_{10}$, respectively. The results from the two AQB systems exhibit slight differences but are consistent overall. Considering both the sensitivity coefficient and hygroscopicity, the performance of AQB in deriving dry PM concentration is significantly improved with lower MAPE, RMSE, and higher $R^2$ than the results obtained using only the sensitivity coefficient, as summarized in Table 2. … The lower $\kappa$ for $PM_{2.5-10}$ might suggest a significant contribution from dust or other less hygroscopic species, consistent with the IC analyses in Table 3 and discussed further in Sect. 3.3. With the retrieved $\alpha$ and $\kappa$ for $PM_{2.5}$ and $PM_{2.5-10}$, Fig. 3(e) shows the scatter distribution between the derived dry $PM_{2.5-10}$ from AQB data and TW-EPA data, exhibiting a MAPE of 31.8%, more significant than the 24.8% for $PM_{2.5}$. … Detection efficiency may be influenced by notable spatial variations, aligning with the findings of Kaliszewski et al. (2020), which showed a reduced correlation between OPC-N3 measurements and reference instruments for larger particles. The dry $PM_{10}$ derived from AQB through the divided $PM_{2.5}$ and $PM_{2.5-10}$ analysis demonstrates better consistency with the reported TW-EPA data than the direct calibration method. This is evidenced by a lower MAPE in Fig. 3(g) (18.2%) compared to Fig. 3(f) (29.2%) and a significant improvement than the simple linear regression method, which has a higher MAPE at 62.5% (Table 2).

This substantiates the importance of considering composition heterogeneity among particle sizes for accurate dry PM derivation. "; and in Section 3.3 (Lines 222-235): "A similar analysis for the winter of 2021 yielded a consistent κ range for PM$_{2.5}$, as illustrated in Fig. S5. This consistency across distinct study periods indicates typical ambient PM$_{2.5}$ hygroscopic characteristics in Kaohsiung City during winter, which can be applied for further discussion with the AQB data. For coarse particles, the more significant variability in κ for PM$_{2.5-10}$ compared to PM$_{2.5}$ can be attributed to the significant fluctuations in the soluble composition of coarse particles, primarily driven by substantial quantities of thenardite (Na$_2$SO$_4$) and halite (NaCl) (Tang et al., 2019). …The derived κ value for PM$_{2.5}$ from IC analysis (0.14-0.27) is consistent with that obtained from AQB analysis (~0.22), while the κ value for PM$_{2.5-10}$ from IC analysis (0.06-0.21) is relatively higher than that from AQB analysis (~0.09) (Table 1 and Fig. 4(a)). The κ differences between the IC and AQB analyses could be attributed to the spatial and temporal variations in aerosols, as well as the different campaign years and locations (~20 km apart, as shown in Fig. S1). These differences might also be influenced by technique uncertainties, such as ammonia and nitrate sampling evaporation during filter sampling (Hering and Cass, 1999; Chen et al., 2021), as well as OPC detection uncertainties and the required parameter assumption in the calculation. Overall, the derived κ values from the OPC data in AQB likely reflect the mean hygroscopicity of both integrated fine and coarse particles. ". Additionally, Fig. S5 is revised as follows:

[Figure]

Figure S5: The hygroscopicity of PM$_{2.5}$ derived from AQB and IC data with an assumed particle density of 1.2 g cm$^{-3}$. The IC_2021 is from 2021 samples collected at the National Kaohsiung University of Science and Technology (22°46'22.4" N 120°24'03.4" E) in Kaohsiung for the period of 8 – 18 December 2021 (diamond: mean value; outliers: < 1st quartile Q1-1.5 interquartile range (IQR) or > 3rd quartile Q3+1.5 IQR).

4. please provide how did you convert ppm to mass for various species?

A: For particulate matter, OPC in the AQB and BAM1020 in the TW-EPA station monitor PM mass concentrations in micrograms per cubic meter (μg/m³); hence, there is no need for unit conversion from ppm to mass concentration. For gaseous species monitored in this study, the concentration is calibrated and expressed in volume mixing ratio as ppm or ppb, the same as EPA data. In our study, conversions from ppm to mass were not performed for gaseous species.

If there is a need for the unit conversion of monitored gas species, it can be calculated using the ideal gas equation as follows:

$$Mass\ concentration = C \times \frac{M_{gas} \cdot P}{R \cdot T}$$

where the unit of *Mass concentration* here uses μg m$^{-3}$ as an example, *C* is the volume mixing ratio (ppmv), $M_{gas}$ is the molecular weight of selected gas (g mole$^{-1}$), *R* is the gas constant (J mole$^{-1}$ K$^{-1}$), *T* is the temperature of air parcel (K), and *P* is the pressure of air parcel (Pa). AQB system monitors meteorological parameters temperature (T) and pressure (P), which can convert the gaseous air pollutant concentration from volume mixing ratio to mass concentration. Furthermore, we replaced "ppm" by "ppmv" and "ppb" by "ppbv" to avoid any potential misunderstanding throughout the whole manuscript.

5. please provide your final conclusions in an itemized list.

A: Thank you for your suggestion to present the final conclusions of our study in an itemized list. In this study, we emphasize our study in three main points:

1. Effectiveness of low-cost systems: The performance of home-built Air Quality Box (AQB) systems was evaluated, demonstrating their effectiveness in capturing meteorological parameters and various pollutant concentrations.
2. Sensitivity analysis and hygroscopicity derivation: PM$_{10}$ was divided into PM$_{2.5}$ and PM$_{2.5-10}$ to account for sensor detection sensitivity and compositional heterogeneity among different particle sizes. With the consideration of sensor sensitivity and hygroscopicity of particles, the derived dry particle mass concentrations showed improved consistency with TW-EPA data than those derived from simple linear regression.
3. Derived hygroscopicity and error discussion: The derived hygroscopicity values align with results from the soluble composition analysis using ion chromatography. This study also emphasizes the need for careful consideration of uncertainties and calibration techniques to interpret low-cost sensor data in atmospheric research accurately.

We prefer to choose the narrative conclusion section by reconstructing the logic in revision: " In this study, we evaluated the performances of home-built Air Quality Box (AQB) systems equipped with low-cost sensors and focused on the ambient variability of particulate matter (PM) concentrations to derive the hygroscopicity of PM and the conversion to dry particle concentrations. The AQB systems revealed their effectiveness in capturing meteorological parameters and most pollutant concentrations with high

correlations (r ≥ 0.96) for temperature, relative humidity, CO, and Ox ($O_3$ + $NO_2$) and moderate correlations (r ≥ 0.48) for NOx and PM, as compared to TW-EPA data. In the PM analysis, $PM_{10}$ was divided into $PM_{2.5}$ and $PM_{2.5-10}$ to account for compositional heterogeneity among different particle sizes. Comparing the AQB-monitored ambient PM data and the TW-EPA data (for dry particles) at RH ≤ 50%, the derived sensitivity coefficients (α) for $PM_{2.5-10}$ (10.58 - 12.37) were higher than those for $PM_{2.5}$ (1.26 - 1.44) likely due to the significant sensitivity variation in the OPC over time. By considering hygroscopicity with the κ-Köhler equation and assuming a constant composition density for sensitivity-corrected AQB data, the derived dry particle mass concentrations show improved consistency with TW-EPA data compared to the simple linear regression approach. The derived κ values range from 0.15 to 0.29 for $PM_{2.5}$ and 0.05 to 0.13 for $PM_{2.5-10}$, consistent with those from IC soluble composition analysis (0.14 to 0.27 for $PM_{2.5}$ and 0.06 to 0.21 for $PM_{2.5-10}$) and primarily influenced by the proportion of soluble components, ~53% in $PM_{2.5}$ and ~30% in $PM_{2.5-10}$. The sensitivity analysis of various parameters showed that the effects of chosen deliquescence relative humidity (DRH) thresholds and Kelvin effects had a minor impact on κ values (less than 1%). Conversely, recalculating particle densities for $PM_{2.5}$ (1.42 ± 0.03 g $cm^{-3}$) and $PM_{2.5-10}$ (1.34 ± 0.07 g $cm^{-3}$) led to higher κ values by approximately 17% and 9%, respectively, compared to the results assuming 1.2 g $cm^{-3}$. Overall, the AQB systems are helpful in understanding the temporal and spatial variability of air quality by effectively monitoring pollutant concentrations and providing the capability for hygroscopicity derivation. This study also emphasizes the need for careful consideration of uncertainties and calibration techniques to accurately interpret low-cost sensor data in atmospheric research."

**RC2**

**We would like to thank the anonymous reviewer for the comments that significantly improve the clarity and readability of the manuscript. Our point-by-point responses are found below in blue ink. The revised content is highlighted in yellow.**

General Comments:

The normalization and portability of atmospheric pollutant monitoring are crucial for in-depth research into local variations in atmospheric environment and pollution. This manuscript establishes a low-cost air quality monitoring device and applies it to estimate the hygroscopicity parameters of aerosols, comparing the results with site observation data. The study holds certain scientific and application value and aligns with the publication scope of the Atmospheric Measurement Techniques journal. However, there are some scientific and technical issues within the manuscript itself, suggesting major revisions before reconsidered.

Specific Comments:

1) The accuracy and error range of AQB in detecting aerosols and the application of AQB to estimate aerosol hygroscopicity parameters should be two separate research components with a sequential order. However, in this manuscript, the authors often fail to clearly distinguish between the two. The authors adopt a method of setting RH thresholds to classify AQB observation results into dry aerosols and humidified aerosols, and then compares and calibrates the observations of dry aerosols with EPA station data, which is a feasible approach. However, in Sections 3.1, 3.2, and Figures 2 and 3, the authors do not classify or analyze the data based on the RH threshold set by themselves. Meanwhile, as shown in Figure 2(b), the occurrence of RH below 50% during the observation period is rare. Can such a limited amount of data support the examination of the reliability of AQB detection?

A: We appreciate the reviewer's insightful comments. In response to concerns, Fig R1, similar in configuration to Figs. 3(a-c), shows the data points at RH $\leq$ 50 %, which were applied to determine the sensitivity coefficient ($\alpha$).

[Figure]

**Figure R1. The correlation of mass concentration between TW-EPA and OPC in AQB #1 (raw data) for PM$_{2.5}$, PM$_{10}$, and PM$_{2.5-10}$ at RH$\leq$ 50%. The hollow points are the two significant outliers mentioned in Table 2.**

Although these data points constitute about 5% of the total (i.e., 17 out of 356 points), a high correlation between the AQB and TW-EPA measurements was observed. Figure 2(b) shows only part of the campaign to reveal the temporal comparison between AQB and TW-EPA data. As stated in section 2.3, the statistical distribution of $M_{EPA}$ to $M_{OPC}$ ratios at RH $\leq$ 50 % was analyzed to evaluate a sensitivity coefficient ($\alpha$) as the mean value $\pm$ 0.5$\sigma$ ($\sigma$: standard deviation). The shaded area in Fig. R1 represents the distribution of $\alpha$, covering most of the data points at RH $\leq$ 50 %. The same calculation but for higher RH thresholds (up to 60% to have more data points, 51 out of 356) summarized below shows a similar $\alpha$ range, indicating sufficient data points at RH $\leq$ 50 % for a conclusive $\alpha$. The following figure is added to the supplementary as Fig. S2 to clarify this issue. In the content, Lines 179-181 were revised as follows: "==The estimated $\alpha$, as summarized in Table 1, are higher for PM$_{10}$ than for PM$_{2.5}$, i.e., 2.02 $\pm$ 0.34 vs 1.26 $\pm$ 0.16, which are reasonably conclusive as tested with more data points selected at higher RH thresholds (Fig. S2).=="

[Figure]

Figure S2: The determined sensitivity as a function of RH thresholds for PM$_{2.5}$ (red), PM$_{10}$ (blue) and PM$_{2.5-10}$ (green). The shading area is the mean value ± 0.5σ

As to the accuracy of applied methods, the comparison analysis between fitted data and TW-EPA data is summarized in a new table (Table 2) added to the content as follows:

[revised manuscript text omitted]

2) Introducing aerosol chemical composition observations into a thermodynamic equilibrium model to calculate aerosol hygroscopic growth and comparing it with optical observations is a common research approach. However, contrasting different field experiments conducted at different times (with an 8-year difference) and different underlying surfaces by the authors doesn't have much significance.

A: Thank you for highlighting the concerns regarding the comparison of derived hygroscopicity between two different field studies conducted in different years. We acknowledge that the hygroscopic characteristics of ambient particles can vary spatially and temporally. Our comparison of the chemical composition of $PM_{2.5}$ from two winter sampling campaigns in 2013 and 2021 revealed that the composition concentration might be different, but the derived hygroscopicity was consistent across these years, demonstrating typical ambient $PM_{2.5}$ hygroscopic characteristics in Kaohsiung City during winter. Even though there was no intensive filter sample collection during the studied period, the derived mean $\kappa$ range under the specific assumption (solute density and ignorance of Kelvin effect) from AQB data was discussed for the range consistency compared to the results of available two winter campaigns for $PM_{2.5}$ and 2013 winter campaign for $PM_{2.5-10}$. However, the temporal resolution for the derived hygroscopicity from IC data is higher than that derived from AQB data since the AQB analysis required a longer time period to cover a comprehensive range of RH to have the particle growth profile. The derived hygroscopicity then represents a mean value over a longer period. To clarify this approach, Section 3.3 (Lines 222-235): "A similar analysis for the winter of 2021 yielded a consistent $\kappa$ range for $PM_{2.5}$, as illustrated in Fig. S5. This consistency across distinct study periods indicates typical ambient $PM_{2.5}$ hygroscopic characteristics in Kaohsiung City during winter, which can be applied for further discussion with the AQB data. For coarse particles, the more significant variability in $\kappa$ for $PM_{2.5-10}$ compared to $PM_{2.5}$ can be attributed to the significant fluctuations in the soluble composition of coarse particles, primarily driven by substantial quantities of thenardite ($Na_2SO_4$) and halite (NaCl) (Tang et al., 2019). …The derived $\kappa$ value for $PM_{2.5}$ from IC analysis (0.14-0.27) is consistent with that obtained from AQB analysis (~0.22), while the $\kappa$ value for $PM_{2.5-10}$ from IC analysis (0.06-0.21) is relatively higher than that from AQB analysis (~0.09) (Table 1 and Fig. 4(a)). The $\kappa$ differences between the IC and AQB analyses could be attributed to the spatial and temporal variations in aerosols, as well as the different campaign years and locations (~20 km apart, as shown in Fig. S1). These differences might also be influenced by technique uncertainties, such as ammonia and nitrate sampling evaporation during filter sampling (Hering and Cass, 1999; Chen et al., 2021), as well as OPC detection uncertainties and the required parameter assumption in the calculation. Overall, the derived $\kappa$ values from the OPC data in AQB likely reflect the mean hygroscopicity of both integrated fine and coarse particles. ". Additionally, the corrected version of Fig. S5 is shown as:

[Figure]

Figure S5: The hygroscopicity of $PM_{2.5}$ derived from AQB and IC data with an assumed particle density of 1.2 g cm$^{-3}$. The IC_2021 is from 2021 samples collected at the National Kaohsiung University of Science and Technology (22°46'22.4" N, 120°24'03.4" E) in Kaohsiung for the period of 8 – 18 December 2021 (diamond: mean value; outliers: < 1st quartile Q1-1.5 interquartile range (IQR) or > 3rd quartile Q3+1.5 IQR).

3) In Table 1, the hygroscopicity parameter kappa for $PM_{2.5}$ and $PM_{2.5-10}$ is smaller than the hygroscopicity parameter kappa for $PM_{10}$, which is abnormal. $PM_{10}$ is the sum of $PM_{2.5}$ and $PM_{2.5-10}$, and its hygroscopicity should be intermediate between the two.

A: The higher hygroscopicity of $PM_{10}$ than those of $PM_{2.5}$ and $PM_{2.5-10}$ is due to a significantly high portion $PM_{2.5}$ in $PM_{10}$ observed in AQB combined with the positive correlation between the derived sensitivity (α) and hygroscopicity (κ), as shown in Eq. (3) for the ambient and dry PM mass concentration conversion.

$$M_{d,derived} = (\alpha \times M_{OPC}) \times \left[\left(\frac{S\kappa}{1-S}\right) \times \frac{\rho_w}{\rho_d} + 1\right]^{-1} \tag{3}$$

For a given $M_{opc}$, a higher α would require a higher κ to have the same $M_{d, derived}$. Because the estimated α for $PM_{10}$ is higher than that for $PM_{2.5}$, a higher κ might be expected if $PM_{2.5}$ portion is dominant in $PM_{10}$. This can be evaluated through the following calculation using an ideal system. The equation in the following shows the derived dry $PM_{10}$ concentration is the sum of the derived $PM_{2.5}$ and the derived $PM_{2.5-10}$ concentration.

$$(\alpha_{10} \times M_{10}) \times \left[\left(\frac{S\kappa_{10}}{1-S}\right) \times \frac{\rho_w}{\rho_d} + 1\right]^{-1}$$
$$= (\alpha_{2.5} \times M_{2.5}) \times \left[\left(\frac{S\kappa_{2.5}}{1-S}\right) \times \frac{\rho_w}{\rho_d} + 1\right]^{-1} + (\alpha_{2.5-10} \times M_{2.5-10}) \times \left[\left(\frac{S\kappa_{2.5-10}}{1-S}\right) \times \frac{\rho_w}{\rho_d} + 1\right]^{-1}$$

where $M$ is $M_{opc}$ for different size ranges (indicated in the subscript). By assuming $X$ as the ratio of $M_{2.5-10}$ to $M_{2.5}$ monitored by AQB, the relationship can be rewritten in the following:

$$\alpha_{10}(1+X) \times \left[\left(\frac{S\kappa_{10}}{1-S}\right) \times \frac{\rho_w}{\rho_d} + 1\right]^{-1} = \alpha_{2.5} \times \left[\left(\frac{S\kappa_{2.5}}{1-S}\right) \times \frac{\rho_w}{\rho_d} + 1\right]^{-1} + (\alpha_{2.5-10} \times X) \times \left[\left(\frac{S\kappa_{2.5-10}}{1-S}\right) \times \frac{\rho_w}{\rho_d} + 1\right]^{-1}$$

where the $\rho_w$ and $\rho_d$ are constants (assumed in this study), the sensitivity coefficients are given as $\alpha_{2.5} = 1.26$, $\alpha_{10} = 2.02$, and $\alpha_{2.5-10} = 12.37$ from Table 1, and hygroscopicity is given as a median value in derived results of $PM_{2.5}$ and $PM_{2.5-10}$ ($\kappa_{2.5} = 0.24$, and $\kappa_{2.5-10} = 0.10$). The derived $\kappa_{10}$ can be evaluated with a function of $X$ and $S$ in the following figure and is mainly affected by $X$. A higher $\kappa_{10}$ than $\kappa_{2.5}$ (0.24) is expected as $X < 0.05$, i.e., low $M_{2.5-10}/M_{2.5}$. The observed data from AQB has a higher probability of having $X \sim 0.04$ with $S \sim 0.8$-$0.9$ which leads to a higher derived $\kappa_{10}$ than $\kappa_{2.5}$ (0.24).

[Figure]

**Figure R2. The distribution of the derived hygroscopicity of $PM_{10}$ in the condition of given sensitivity coefficients ($\alpha_{2.5} = 1.26$, $\alpha_{10} = 2.02$, and $\alpha_{2.5-10} = 12.37$) and hygroscopicity ($\kappa_{2.5} = 0.24$, and $\kappa_{2.5-10} = 0.10$). The contour is the derived hygroscopicity of $PM_{10}$, and the shading is the data point distribution of the AQB monitored data.**

Overall, the estimated higher derived $\kappa_{10}$ is possible to happen due to a significant amount of data points having a low portion $PM_{2.5-10}$ in $PM_{10}$ observed in AQB.

4) In Section 2.1, the author introduces the photo-ionization detector for monitoring VOCs, but in Figure 1 and subsequent manuscripts, the abbreviation used by the author is NMHC. These two abbreviations are not entirely equivalent.

The Alphasense PID-AH2 measures volatile organic compounds (VOCs) in the air using photoionization detection (PID), as stated in its datasheet. This device utilizes a lamp that emits high-energy UV photons. When a VOC molecule absorbs a photon, it generates electrically charged ions, creating an electric field. The detector then monitors the resulting current, which is proportional to the ambient VOC concentration. Notably, the PID-AH2 in this study uses a Krypton lamp with a photon energy of about 10.6 eV, capable of detecting some C2, and most C3, C4+ VOCs. In contrast, the Horiba APHA-360, a VOC gas analyzer used by TW-EPA, continuously analyzes THC, $CH_4$, and non-$CH_4$ (NMHC) in ambient air using a flame ionization detector and cross flow modulation. Since the

ionization potential of methane is approximately 13.7 eV, the PID cannot detect its concentration, making NMHC the closest comparable data from TW-EPA for our purposes. Additionally, the sensitivity of the PID varies with the type of VOC detected; for example, toluene generates approximately twice the response of isobutylene. Consequently, as shown in Fig. 2(g), the sensor captures some peak NMHC concentrations, but not all temporal variations are detectable. To ensure consistency, the label in Fig.1 is revised as "VOCs". Furthermore, to avoid any potential confusion regarding the capabilities of the PID sensor, the following information is added in Section 2.1 (Lines 71-73): " The PID sensor, equipped with a Krypton lamp providing a photon energy of about 10.6 eV, cannot detect methane, which has a higher ionization potential of ~13.7 eV (Glockler, 1926). Therefore, the data of non-methane hydrocarbons (NMHC) from TW-EPA is more comparable to PID data in our analysis."

[Figure]

Figure 1: The design of the AQB system.

---

## Author Response (AR2)

**We would like to thank the editor and anonymous reviewers for the comments that significantly improved the clarity and readability of the manuscript. Our point-by-point responses are found below in blue ink. The revised content is highlighted in yellow.**

**Part1.**

**Responses to editor comments:**

Public justification (visible to the public if the article is accepted and published):

This manuscript presents a novel attempt to use low-cost sensors to determine particle hygroscopicity. It requires an additional minor review to address remaining comments by both the reviewers and the editor.

Additional private note (visible to authors and reviewers only):

1. Both reviewers have additional comments that I think need to be addressed by the authors. Referee #2 would like to see further discussion of errors in the calculated hygroscopicity.

A: Please find our response to RC2 listed in part 2.

2. Referee #3 has several comments, including a better discussion of the setup of the AQB.

A: Please find our point-by-point responses to RC3 (Q1-Q3) listed in part 3.

3. Referee #3 also notes that parts of the manuscript are difficult to follow.

A: With the comment (Q5) from Referee #3, the content has been revised. The results and discussion section was re-arranged to improve the logicality and clarification as follows (the content was revised but not shown here): "

3 Results and Discussion
    3.1 Performance of AQB systems
    3.2 Comparison between OPC and BAM data
        3.2.1 Sensitivity coefficient of OPC
        3.2.2 Hygroscopicity derivation
    3.3 Hygroscopicity derivation using IC data
        3.3.1 Composition analysis
        3.3.2 E-AIM analysis
    3.4 Sensitivity of assumed parameters on derived hygroscopicity"

4. I would like to echo Referee #3's comments and add several of my own. This manuscript has many abbreviations and acronyms, which can make it hard to follow. A glossary would be helpful. At a minimum, the authors need to define each abbreviation or acronym the first time that it it appears in the manuscript. For example, AQB appears in the abstract but needs to be defined again in the main text.

A: Thank you for your suggestion. The content has been revised to ensure that all abbreviations are introduced after their full names when they appear for the first time in the main content. Additionally, a list of principle symbols and abbreviations is provided in Lines 24-60 for clarification.

5. I also share Referee #3's confusion about how the OPC and regulatory data are compared. The manuscript briefly describes a calibration for the OPC from co-location with the BAM (section 2.2). The subsequent hygroscopicity analysis seems to use the calibrated OPC data and compares it to the (dry) PM measured by the BAM. Thus, I don't understand how the comparison between the OPC and BAM can be used both for calibration and the calculation of kappa. The authors should clarify exactly what process they are using at each step.

A: To clarify the calibration process, a detailed description was added to supplementary material as follows:

**"Aerosol hygroscopicity derived using OPC and BAM data**

The optical particle counter (OPC) (model: OPC-N2, Alphasense) provides digital outputs of $PM_1$, $PM_{2.5}$, $PM_{10}$, and optionally $PM_{4.25}$, along with histograms of the particle counts for 16 size bins ranging from 0.38 to 17 μm. This device is designed to monitor ambient aerosol concentrations without any drying system attached to the sampling inlet. In contrast, the Beta Attenuation Mass Monitor (BAM) (model: BAM1020, Met One Instrument) is designed to monitor dry particle mass concentrations of $PM_{2.5}$ and $PM_{10}$, using a heating device to ensure the sampling relative humidity (RH) remains below 50%. If the RH of the sampled air stream exceeds 50%, the inlet heater activates, reducing the RH to approximately 35% by warming the air stream downstream before reaching the filter tape. If the RH is below 50%, the heater remains inactive, not altering the sampling flow RH. The technical specifications for the OPC and BAM are summarized in Table S4.

To derive aerosol hygroscopicity (κ), the sensitivity coefficient of OPC was evaluated first using the data points at RH ≤ 50%, as described in Section 2.3 of the main content. Depending on the size range, $\alpha_{2.5}$, $\alpha_{10}$, and $\alpha_{2.5-10}$ represent the sensitivity coefficient of OPC for $PM_{2.5}$, $PM_{10}$, and $PM_{2.5-10}$, respectively. For $PM_{10}$, a range of hygroscopicity of 0 to 1.2 was applied to Eq. (S1) to obtain $M_{d,derived,10}$ and evaluate the mean absolute percentage error (MAPE) between $M_{d,derived,10}$ and $M_{BAM,10}$. MAPE as a function of the applied hygroscopicity is plotted to determine the $\kappa_{10}$ range, which has MAPE ≤ 1.1×the lowest MAPE, considering the uncertainty. A similar calculation is applied to $PM_{2.5}$, and $PM_{2.5-10}$ to derive $\kappa_{2.5}$ and $\kappa_{2.5-10}$, directly.

$$M_{d,derived,10} = \alpha_{10} \times M_{OPC,10} \times \left[ \left( \frac{S\,\kappa_{10}}{1-S} \right) \times \frac{\rho_w}{\rho_d} + 1 \right]^{-1} \tag{S1}$$

However, $PM_{10}^*$ in Table 1 considers the size-dependent sensitivity with a mean hygroscopicity. $\kappa_{10}$ is derived using the following equation:

$$M_{d,derived,10} = \left( \alpha_{2.5} \times M_{OPC,2.5} + \alpha_{2.5-10} \times M_{OPC,2.5-10} \right) \times \left[ \left( \frac{S\,\kappa_{10}}{1-S} \right) \times \frac{\rho_w}{\rho_d} + 1 \right]^{-1} \tag{S2}$$

The derived $\kappa_{10}$ using Eq. (S2) ranges from 0.13 to 0.23, falling between κ values for $PM_{2.5}$ and $PM_{2.5-10}$, and is more reasonable than from Eq. (S1), as discussed in Section 3.2.2."

**Part 2.**

**Responses to referee#2's comments:**

1. The authors have provided a detailed point-by-point response to the previous round of review comments, resolving most of the issues. Regarding specific comment (3), the authors' response explained the reason why the kappa value of $PM_{10}$ is higher than those of $PM_{2.5}$ and $PM_{2.5-10}$. However, this explanation also indicates that there is a significant error in estimating aerosol hygroscopicity using AQB detection results, as the estimation does not align with the real atmospheric conditions. This error might originate from the process of calculating the derived sensitivity, such as whether the 50% threshold is too high, or it could be due to the derived sensitivity under dry conditions not being well applicable to humid conditions. The authors should thoroughly analyze the causes of this error and its magnitude in the manuscript.

A: The higher kappa value for $PM_{10}$ compared to $PM_{2.5}$ and $PM_{2.5-10}$ in Table 1 is likely due to significant sensitivity difference between $PM_{2.5}$ and $PM_{2.5-10}$ in our system and the simplified calculation using one sensitivity coefficient for the overall $PM_{10}$. $\kappa$ for $PM_{10}$ ($\kappa_{10}$) is derived using the following equation:

$$M_{d,derived,10} = \alpha_{10} \times M_{OPC,10} \times \left[\left(\frac{S\,\kappa_{10}}{1-S}\right) \times \frac{\rho_w}{\rho_d} + 1\right]^{-1} \tag{R1}$$

where the parameters with subscript 10 stand for $PM_{10}$. As $PM_{10}$ is divided into $PM_{2.5}$ and $PM_{2.5-10}$, the different estimated sensitivity coefficients suggest that the response of the applied OPC is size-dependent. By applying the more sophisticated sensitivity correction, $(\alpha_{2.5} \times M_{OPC,2.5} + \alpha_{2.5-10} \times M_{OPC,2.5-10})$ represents the ambient $PM_{10}$ concentration. The single hygroscopicity parameter of $PM_{10}$ can be estimated using the following equation:

$$M_{d,derived,10} = (\alpha_{2.5} \times M_{OPC,2.5} + \alpha_{2.5-10} \times M_{OPC,2.5-10}) \times \left[\left(\frac{S\,\kappa_{10}}{1-S}\right) \times \frac{\rho_w}{\rho_d} + 1\right]^{-1} \tag{R2}$$

The derived kappa using Eq. (R2) ranges from 0.13 to 0.23, falling between $\kappa$ values for $PM_{2.5}$ and $PM_{2.5-10}$. However, because aerosols are inhomogeneous among different particle sizes, we focus on addressing different hygroscopicities between fine and coarse particles and verifying the retrieved values with other methods. OPC has become an affordable aerosol detector and is widely applied for environment monitoring. If the sensitivity is the same across different particle sizes, Eq. (R1) would be sufficient for related calculations. Otherwise, detailed analyses with separated size ranges would provide more comprehensive results. A more detailed deriving process was added to the supplementary material.

Table 1 is revised to include the results retrieved using Eq. (R2) as follows:

**Table 1: The sensitivity coefficients and the hygroscopicity for $PM_{2.5}$, $PM_{10}$ and $PM_{2.5-10}$.**

|  | Sensitivity coefficient ($\alpha$) | | Hygroscopicity ($\kappa$) | | | |
|---|---|---|---|---|---|---|
|  | AQB #1 | AQB #2* | AQB #1 | AQB #2 | IC (species) | IC (E-AIM) |
| $PM_{2.5}$ | $1.26 \pm 0.16$ | $1.44 \pm 0.20$ | $0.18 - 0.29$ | $0.15 - 0.24$ | $0.14 - 0.27$ | $0.14 - 0.26$ |
| $PM_{10}$ | $2.02 \pm 0.34$ | $2.20 \pm 0.38$ | $0.20 - 0.39$ | $0.18 - 0.30$ |  |  |
| $PM_{10}^{\#}$ | $\alpha_{2.5}, \alpha_{2.5-10}$ | $\alpha_{2.5}, \alpha_{2.5-10}$ | $0.13 - 0.23$ | $0.11 - 0.26$ |  |  |
| $PM_{2.5-10}$ | $12.37 \pm 1.33$ | $10.58 \pm 2.90$ | $0.07 - 0.13$ | $0.05 - 0.09$ | $0.06 - 0.21$ | $0.08 - 0.21$ |

**the hygroscopicity derived using different sensitivity coefficients for different size ranges. $\alpha_{2.5}$ and $\alpha_{2.5-10}$ are sensitivity coefficients for $PM_{2.5}$ and $PM_{2.5-10}$, respectively. More details are provided in the description of the supplementary material.**
* the sensitivity of AQB #2 presents the value in the period of sampling flow rates at 3.6-4.2 LPM

The content of section 3.2.2 (Lines 254-260) is revised to include this concept as follows:

"The dry $PM_{10}$ derived from OPC through the divided $PM_{2.5}$ and $PM_{2.5-10}$ analysis demonstrates better consistency with the reported BAM data than the direct calibration method. This is evidenced by a lower MAPE in Fig. 3(g) (18.2%) compared to Fig. 3(f) (29.2%) and a significant improvement than the simple linear regression method, which has a higher MAPE at 62.5% (Table 2). Moreover, the derived $\kappa$ for $PM_{10}$ with the size-dependent sensitivity coefficient correction ranges from 0.13 to 0.23 (Table 1). This value falls between those for $PM_{2.5}$ and $PM_{2.5-10}$ and is more reasonable compared to $\kappa$ derived with a fixed sensitivity coefficient ($\kappa = 0.20$-$0.39$, higher than those for $PM_{2.5}$ and $PM_{2.5-10}$). The results substantiate the importance of considering composition heterogeneity among particle sizes for accurate dry PM derivation. "

**Part 3.**

**Responses to referee#3's comments:**

The submitted manuscript presents results of hygroscopicity measurements using an inexpensive set of aerosol analyzers. The work comprehensively describes the measurement results and is very valuable. However, there are several issues that need clarification and elaboration.

Specific Comments:

1. The manuscript lacks a more detailed description of the measurement system. Although the authors presented a photograph of the interior of the setup and a brief description of the devices, there is a lack of description of the sampling system. Did the devices independently sample the measured air from their own inlets, or were they equipped with a common measurement line? I am unable to assess this because no photograph of the setup in its operational state was presented. Furthermore, the placement of the device in relation to the inlet of the reference measurement station is crucial. The effects of boundary layer and local turbulent flows in urbanized areas can significantly influence aerosol measurement results. I believe this should be discussed in greater detail.

A: The home-built Air Quality Box (AQB) system samples the air independently with a fan visible in the upper left of Fig. 1. Additionally, the optical particle counter (OPC-N2) has its own fan ambient air sampling. On the opposite side of the sampling fan, AQB has two openings for air to flow out. The selected TW-EPA station is situated on the roof of a building in a well-ventilated environment. Because the TW-EPA station has a standard operation process to ensure consistency between stations, our system can only be set aside the station (approximately 5 m horizontally and 2 m vertically from the sampling inlet of TW-EPA station) with a photo shown in Figure S1(b). The content of Section 2.1 (Lines 117-121) was revised to describe more detail of the AQB system as follows:
"The entire system is housed in a remodeled enclosure with a dimension of 25 cm × 16 cm × 8 cm (L × D × H) and has well-ventilated openings for sampling and exhaust. The sampling flow rate is primarily controlled by an installed fan at ~ 5.6 L min$^{-1}$, corresponding to a residence time of approximately 34 s in the box. This configuration allows the system to effectively monitor ambient air quality independently without the need of additional inlets."

The other revised paragraph is shown in Section 2.2 (Lines 123-126) as follows:
" The calibration of AQB sensors was carried out by co-locating them with TW-EPA Nanzi station (Fig. S1) in Kaohsiung, Taiwan (22°44'12" N, 120°19'42" E) from 4 to 19 February 2021. Nanzi station is situated on the roof of a 15 m high building in a well-ventilated environment. The primary gaseous components, dry $PM_{2.5}$ and $PM_{10}$ concentrations, and basic meteorological parameters are continuously monitored using standard instruments, as summarized in Table S1."

Additionally, Fig. 1 was revised to include the information of fans to emphasize the sampling flow as follows:

[Figure]

Figure 1: The design of the air quality box (AQB) system.

For the influence of boundary layer and local turbulent flows, the performance of CO monitoring exhibits a high correlation (r=0.976) between AQB and TW-EPA, as depicted in Fig. 2(c), indicating that the two systems are co-located in a similar environment. Therefore, the effects of boundary layer and local turbulent flows in urbanized areas can be assumed negligible for the comparision between OPC and BAM data. The content of section 3.1 (Lines 199-202) was revised as follows:

"Figure 2 shows the time series of the meteorological parameters and pollutant concentrations between calibrated AQB and TW-EPA data from 14 to 17 February 2021. T, RH, CO, and Ox showed a good correlation with r > 0.9, while NO, $NO_2$, $PM_{2.5}$, and $PM_{10}$ had a moderate correlation (r ≥ 0.48). The high correlation (r=0.976) for CO (with a lifetime of ~ 2 months) indicates a similar air parcel sampled by both AQB and the instrumentation in TW-EPA."

Moreover, Fig. S1 is revised with the photograph of the setup condition of AQB system co-located with TW-EPA station as follows:

[Figure]

**Figure S1. (a) Location of TW-EPA Nanzi station (AQB calibration campaign) site and Fooyin University (2013 sampling campaign). (from © Google Earth 2024 and © Google Maps 2024). (b) Photograph of the setup in AQB system operational state. AQB was located approximately 5 m horizontally and 2 m vertically from the sampling inlet of TW-EPA Nanzi station).**

2. Additionally, the differences between AQB and EPA are not clear to me. It would be beneficial to at least discuss the differences in resolution between the two measurement systems (e.g., in the supplement). The devices used at the reference station should also be calibrated, but how was this done?

A: Table S1 was revised to include more detailed information, including the detection range and resolution for sensors in AQB, and intruments at TW-EPA as follows:

**Table S1: Summary of the applied sensors in AQB and instruments at TW-EPA station.**

| | Sensors in AQB (Manufacturer) | Detection range, Detection resolution | Instruments at TW-EPA (Manufacturer) | Detection range, Detection resolution |
|---|---|---|---|---|
| T, RH | SHT31 (Seeed) | T: -40 – 125 °C, 0.1 °C
RH: 0 – 100%, 0.1% | 083D (Met One Instruments) | T: -30–50 °C, 0.1 °C
RH: 0–100%, 0.04% |
| CO | CO-B4 (Alphasense) | 0–1000 ppmv, in ppbv | APMA360 (Horiba) | 0–100 ppmv, 0.02 ppmv |
| NO | NO-B4 (Alphasense) | 0–20 ppmv, in ppbv | ML9841 (Horiba) | 0–20 ppmv, 1 pptv |
| $NO_2$ | NO2-B43F (Alphasense) | 0–20 ppmv, in ppbv | ML9841 (Horiba) | 0–20 ppmv, 1 pptv |
| $O_3$ | OX-B431 for $O_3$+$NO_2$ (Alphasense) | 0–20 ppmv, in ppbv | ML9810 (Ecotech) | 0–20 ppmv, 1 pptv |
| $SO_2$ | SO2-B4 (Alphasense) | 0–100 ppmv, in ppbv | ML9850 (Ecotech) | 0–20 ppmv, 1 pptv |
| VOC | PID-AH2 (Alphasense) | 0–40 ppmv, in ppbv | APHA360 (Horiba) | 0–100 ppmv, 0.022 ppmv |
| PM | OPC-N2 (Alphasense) | 0.01–1500 µg m$^{-3}$, 0.1µg m$^{-3}$ | BAM1020 (Met One Instruments) | 0–10,000 µg m$^{-3}$, 0.1µg m$^{-3}$ |

Table S1 shows that the instruments in TW-EPA have a higher detection resolution, making them suitable reference devices for calibrating low-cost sensors. In this study, we focus on deriving aerosol hygroscopicity using two PM instruments, OPC and BAM. Detailed information was added to Table S4 as follows:

**Table S4: Technical specifications for OPC and BAM**

| | OPC | BAM |
|---|---|---|
| Manufacturer | Alphasense | Met One Instruments |
| Model | OPC-N2 | BAM1020 |
| Particle range (µm) | 0.38 – 17 | - |
| Bin number | 16 | - |
| Laser wavelength (nm) | 658 | - |
| Refractive index | 1.5 + 0i | - |
| Setting particle density (g cm$^{-3}$) | 1.65 | - |

| | | |
|---|---|---|
| Max particle count rate (s$^{-1}$) | 10,000 | - |
| Detection range ($\mu$g m$^{-3}$) | 0.01-1500 (for PM$_{10}$) | 0 – 10,000 |
| Measurement resolution ($\mu$g m$^{-3}$) | 0.1 | 0.1 |
| Sampling flow rate (LPM) | ~5 | 16.67 |
| Coincidence probability (% at 10$^6$ L$^{-1}$) | 0.84 | - |
| Unit dimensions (L × D × H) | 75 × 60 × 63.5 (mm) | 36.2 × 43.2 × 46.7 (cm) |
| Weight | 105 g | 19 kg |
| Particle size designations ($\mu$m) | - | PM$_{10}$, PM$_{2.5}$, and PM$_{2.5-10}$ |
| Filter tape | - | Continuous glass fiber filter |

As to the calibration process for the reference station, the TW-EPA station operates as a standard air quality monitoring station following standard operation procedures of the U.S. Environmental Protection Agency (EPA). The air monitoring analyzers are equipped to perform automatic zero and span calibrations periodically, making self-adjustments to predetermined readings every midnight. Annual multi-point verification/calibrations are conducted to confirm the linearity and calibration slope of the selected calibration scale. For the PM monitoring instrument, the detection sensitivity, accurate PM$_{2.5}$ cut point, temperature, barometric pressure sensors, and flow rate device undergo annual calibration. More detailed information can be found in the document "Quality Assurance Handbook for Air Pollution Measurement Systems" (https://www.epa.gov/sites/default/files/2020-10/documents/final_handbook_document_1_17.pdf)

3. In the description and discussion of the results, I missed more concrete references to the measurement results of other authors. While the processes behind the described results are explained, there is a lack of references to other literature values.

A: Thank you for your kind reminder. The measurement results from other authors in reference were added to the results and discussion context. The content of Sect. 3.2.1 (Lines 225-231) were revised as follows: "With sensitivity calibration, the performance at ambient RH ≤ 50% exhibits a strong correlation with MAPE at 12.8%, 18.5%, and root mean squared error (RMSE) at 3.7 $\mu$g m$^{-3}$, 10.3 $\mu$g m$^{-3}$ for PM$_{2.5}$ and PM$_{10}$, respectively, as summarized in Table 2 excluding the two significant outliers (shown as hollow circles in Fig. 3). The results confirm the effectiveness of OPCs in capturing PM concentrations after proper calibration, consistent with other real-time outdoor field studies, reporting R$^2$ ranging from 0.34 to 0.97, RMSE ranging from 0.52 to 12.3 $\mu$g m$^{-3}$, and MAPE about 22% (Gillooly et al., 2019; Demanega et al., 2021; Sá et al., 2022; Crilley et al., 2018)." The content of Sect. 3.2.2 (Line 250-254) were revised as follows: "The higher MAPE might result from the low particle number concentration in the coarse mode, with only about 0.01 to 0.1 particles per bin cm$^{-3}$ in the size range of 3.0 to 10.0 $\mu$m. The results are consistent with the findings of Kaliszewski et al. (2020), which showed that the correlation between OPC-N3 (a newer version of OPC-N2) and AeroTrak 8220 (TSI INC., Shoreview, MN, USA) measurement data decreases with particle size, from 0.3-0.5 $\mu$m (r=0.99) to 5-10 $\mu$m (r=0.74).". The additional discussion was added on Sect. 3.2.2 (Line 241-244) as follows:" With a similar methodology, Crilley et al. (2018) applied $\kappa$-Köhler equation to compare measured data between OPC-N2 and tapered element oscillating microbalance (TEOM) and derived the $\kappa$ ranging from 0.38 to 0.41 and 0.48 to 0.51 for PM$_{2.5}$ and PM$_{10}$, respectively, which is within the range for Europe (i.e., 0.36±0.16) (Pringle et al., 2010)."

4. Inexpensive aerosol analyzers were created particularly to enable long-term observations. It seems to me that the time series used is relatively short. I am not saying it is insufficient, but more extensive

measurements could improve the statistics of the presented results. For example, why did the authors choose to conduct measurements in February? What impact could this have had on the measurements?

A: In this study, we selected the winter season as our research period because air pollutant concentrations are typically higher in Kaohsiung City during winter than in summer. The summer months often experience significant rainfall, including afternoon thunderstorms and typhoons, which reduce PM concentrations through wet deposition. While we agree that more extensive measurements could improve the results, it is important to note that aerosol composition varies seasonally in Kaohsiung City, influenced by monsoonal climates and local circulation patterns. Long-term monitoring data spanning multiple seasons might reveal different aerosol characteristics.

Our monitoring campaign began in early February, transitioning into the spring rainy season in March. Moreover, the availability of our measurement at the studied TW-EPA station was limited, so only two weeks of observation were performed for this study. However, the data covered a wide range of RH for the hygroscopicity derivation. Consequently, the hygroscopicity derived from the collected data can be applied for comparison with that calculated from ion chromatography (IC) analysis during the winter campaigns.

5. Some sections, such as 3.3 and 3.4, are written in a single stream of thought, without a clear division into logical parts in the form of paragraphs. This makes these essentially crucial parts of the work difficult to read and may cause problems for readers in understanding the authors.

A: Thank you for your suggestion. The results and discussion have been re-arranged and revised to improve the logicality and clarification as follows (the content was revised but not shown here): "

3 Results and Discussion
    3.1 Performance of AQB systems
    3.2 Comparison between OPC and BAM data
        3.2.1 Sensitivity coefficient of OPC
        3.2.2 Hygroscopicity derivation
    3.3 Hygroscopicity derivation using IC data
        3.3.1 Composition analysis
        3.3.2 E-AIM analysis
    3.4 Sensitivity of assumed parameters on derived hygroscopicity"

Detailed observations:

6. In the text, the parameter k appears earlier (line 52) than its name is introduced (line 104).

A: The content has been revised to ensure that all abbreviations are introduced after their full names when it appears first time in the main content. The content of Lines 87-89 was revised as "Notably, Crilley et al. (2018) improved OPC mass concentration correction by applying the derived hygroscopicity ($\kappa$) values of 0.38-0.41 and 0.48-0.51 for $PM_{2.5}$ and $PM_{10}$, respectively, achieving a 33% improvement." Additionally, a list of principle symbols and abbreviations is provided in Lines 24-60 for clarification.

7. Please check if all elements of the equations are described. For example, what is Sk in equation (3)?

A: Thank you for your kind reminder. The product of two parameters is revised with adding space. The description of elements is also checked and revised. Eq. (2) is revised as follows:"

$$S = \frac{D_{amb}^3 - D_d^3}{D_{amb}^3 - D_d^3(1-\kappa)} exp(\frac{4\sigma_{s/a} M_w}{R\,T\,\rho_w D_{amb}}) \qquad (2)$$

where $D_{amb}$ and $D_d$ are the diameters (m) of the ambient and dry particulate matter, respectively, $\sigma_{s/a}$ is the surface tension of the particle (J m$^{-2}$), $M_w$ is the molecular weight of water (g mole$^{-1}$), $R$ is the gas constant (J mole-1 K$^{-1}$), $T$ is the temperature, and $\rho$w is the density of liquid water (1.0 g cm$^{-3}$)."

For Eq. (3), S$\kappa$ is revised as "S $\kappa$" represents the product of water saturation ratio (S) and hygroscopicity ($\kappa$) in the numerator. To avoid the misleading, Eq. (3) is revised as follows:

$$M_{d,derived} = (\alpha \times M_{OPC}) \times \left[ \left(\frac{S\,\kappa}{1-S}\right) \times \frac{\rho_w}{\rho_d} + 1 \right]^{-1} \qquad (3)$$

8. In section 2.4, the authors describe the various instruments used in the alternative method. Is this method more important? The devices used in this analysis are listed by name in the main text, while the names of the devices used in AQB and EPA are presented in the supplement. Does this mean they are less important? I sense a lack of consistency from the authors in the description of the equipment used.

A: The methodology described in section 2.4 (IC analysis), which requires aerosol sampling, composition analysis, and the calculation of hygroscopic characteristics, represents a typical approach to assessing the hygroscopicity of aerosols through composition analysis. As low-cost sensors are applied to increase the spatial and temporal measurements, a scientific calibration method might be helpful to provide comparison results to the standard instruments. The hygroscopicity parameter is an important factor in the conversion of ambient particles to dry particles. Even though it is less labor-intensive, the data acquisition requires a broader RH coverage. The report hygroscopicity is a mean value over the studied period. The resolution using composition analysis is higher depending on the sampling period, usually half a day. We agree that referring to the devices as AQB and EPA could diminish their representativeness. Therefore, we have focused our analysis on the comparison between the Optical Particle Counter (OPC) and the Beta Attenuation Mass Monitor (BAM). We reviewed all content for PM-related analysis and discussion and replaced AQB with OPC and TW-EPA with BAM. The content of Section 2.2 (Lines 130-134) was revised as follows:
"For PM, the reported values by Beta Attenuation Mass Monitor (BAM) in the TW-EPA station reflect the dry-state PM concentration by controlling the measurement at RH less than 50% (i.e., a heating device applied to reduce the sampling flow to 35% water saturation when the ambient RH is > 50%). On the contrary, the optical particle counter (OPC) in AQB directly monitors ambient PM concentration. The difference between BAM and OPC data reflects the amount of liquid water content in ambient conditions."

9. I believe the manuscript is worth publishing; however, the authors need to work on the readability and logical flow of their work.

A: We appreciate the constructive comments, which have helped us improve the clarity and logical flow of our manuscript.